# Translating genomic tools to Raman spectroscopy analysis enables high-dimensional tissue characterization on molecular resolution

Manuel Sigle [1], Anne-Katrin Rohlfing [1], Martin Kenny[2,3], Sophia Scheuermann [4,5], Na Sun[6], Ulla Graeßner[4], Verena Haug[1], Jessica Sudmann [1], Christian M. Seitz [4,5], David Heinzmann[1], Katja Schenke-Layland [5,7,8], Patricia B. Maguire[2,3,9], Axel Walch[6], Julia Marzi [5,7,8,10] & Meinrad Paul Gawaz [1,10] ✉

Spatial transcriptomics of histological sections have revolutionized research in life sciences and enabled unprecedented insights into genetic processes involved in tissue reorganization. However, in contrast to genomic analysis, the actual biomolecular composition of the sample has fallen behind, leaving a gap of potentially highly valuable information. Raman microspectroscopy provides untargeted spatiomolecular information at high resolution, capable of filling this gap. In this study we demonstrate spatially resolved Raman "spectromics" to reveal homogeneity, heterogeneity and dynamics of cell matrix on molecular levels by repurposing state-of-the-art bioinformatic analysis tools commonly used for transcriptomic analyses. By exploring sections of murine myocardial infarction and cardiac hypertrophy, we identify myocardial subclusters when spatially approaching the pathology, and define the surrounding metabolic and cellular (immune-) landscape. Our innovative, label-free, non-invasive "spectromics" approach could therefore open perspectives for a profound characterization of histological samples, while additionally allowing the combination with consecutive downstream analyses of the very same specimen.

The 'omics revolution has profoundly changed our ability to characterize cells[1]. In 2020, *Nature Methods* has crowned spatially resolved transcriptomics as Method of the Year[2]. This method puts single-cell RNA data in spatial context, providing a powerful tool for a fundamental understanding of the underlying biology and disease mechanisms. However, a comprehensive assessment of these mechanisms requires the integration of information from a large number of diverse data. In spatial transcriptomics, this understanding exceptionally relies on information about how gene activity orchestrates complex cellular arrangements[1].

Thus, the concept of transcriptomic analysis brings into focus, what will be there in future once translated – neglecting the current biochemical constitution of the sample. In addition, directional blood flow, gradients of signal molecules and metabolites, secretion and absorption of biomaterials generate a high-complex dynamic microenvironment, encompassing not only cells but also extracellular matrix, which is not considered in transcriptomics.

To overcome these limitations, we developed a strategy to transfer the advantages of sophisticated analytical tools for transcriptome analyses to another context, the "Raman world". Raman

spectroscopy provides insights into the chemical make-up of a sample, by shining a laser on a sample and measuring the induced scattered photons of a different energy to the incident photons[3] (Fig. 1a). This shift in energy (Raman shift) is indicative of discrete vibrational modes of polarizable molecules, and thus a qualitative measurement of the biochemical composition of the tissue can be obtained[4]. By incorporating intensities, quantitative information can also be inferred. Consequently, a distinctive biological 'fingerprint' can be derived from biological samples, which in turn can be used to detect endogenous macromolecules, metabolites, extracellular matrix, cell types etc. in a non-invasive, label-free manner[4–6].

Although Raman microspectroscopy has been traditionally used for analytical chemistry applications, there has been a notable rise in the use of this technique within biological studies, particularly in the field of biomedicine[4,7,8]. In cardiovascular research, Raman microspectroscopy has been used to detect cardiac biomarkers[9], to evaluate different stages of myocardial infarction and its repair[10,11], to study redox state of cytochromes under ischemic conditions of acute infarction[12] or to monitor collagen remodeling in cardiac tissue engineering[13,14]. In a recent study, Raman spectra indicated a conformational change, or different degree of phosphorylation/methylation, in tyrosine-rich proteins of failing hearts in a rat heart failure model[15]. However, the underlying mechanisms of myocardial

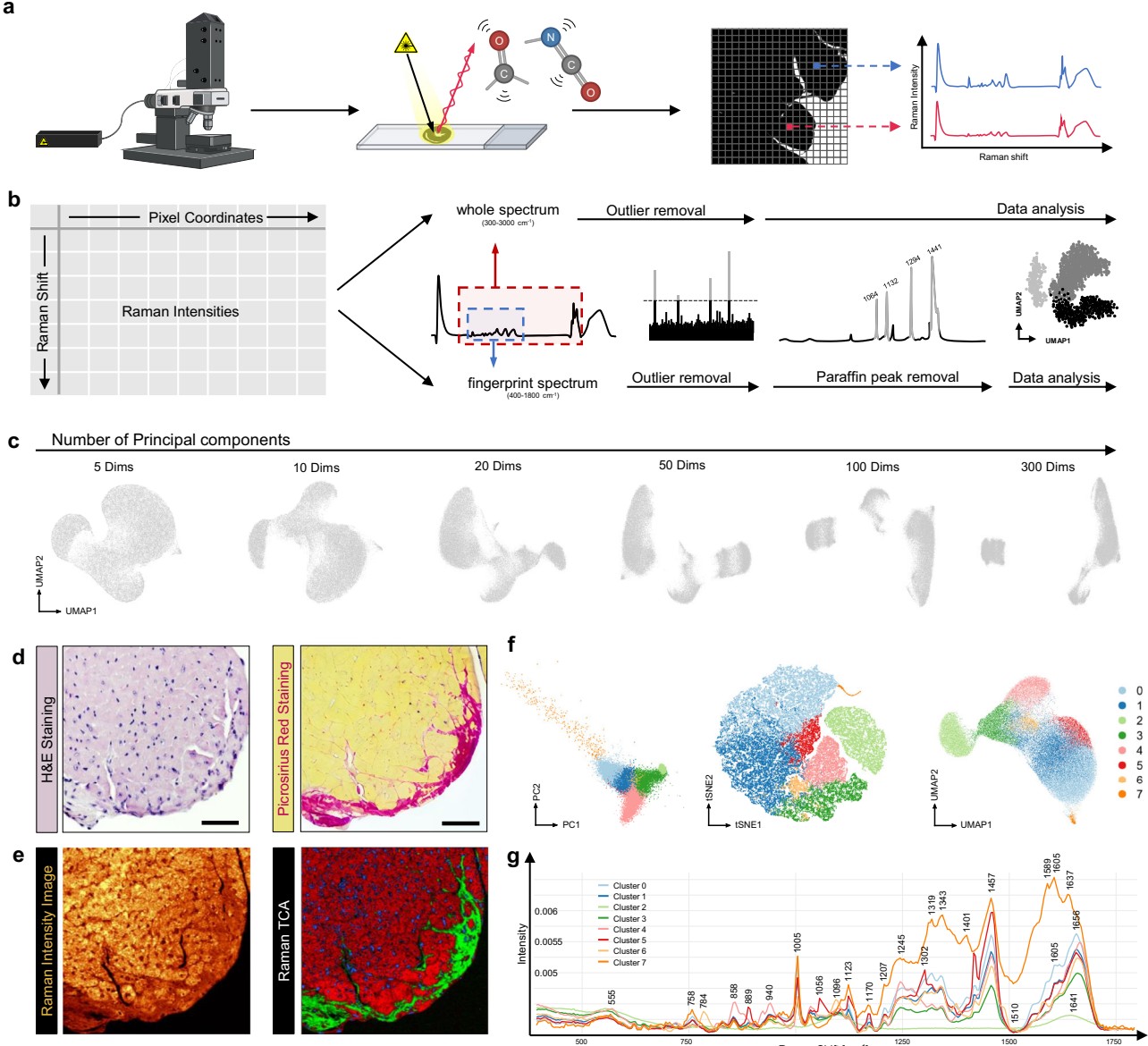

**Fig. 1 | Spatially-unaware spectra from Raman analysis show distinct clustering. a** Data analysis workflow, starting from data acquisition using a confocal Raman microspectroscope. **b** Data handling, transformation and correction. Biologically relevant spectrum between 300 and 3000 cm⁻¹ (named "whole spectrum") was corrected for outliers and then analyzed. Raman "fingerprint" spectrum between 400 and 1800 cm⁻¹ was corrected for outliers and paraffin peaks and then analyzed. **c** Increasing the number of principal components concretizes cluster results for UMAP analysis of Raman data. Whole spectrum data from the section from **d** and **e** was used. **d** H&E (left) and Picrosirius Red Staining (right) of adjacent sections to the sample of subendocardial fibrosis which was subjected to Raman spectroscopy. Scale bar 50 μm. **e** Raman Intensity Image (left) and Raman TCA (right) as described in the Methods section. **f** Various projections after dimensionality reduction of the fingerprint spectrum from the section of **d** and **e**, including PCA, tSNE and UMAP. **g** Cluster-specific average spectra with characteristic peak annotations. UMAP: Uniform Manifold Approximation and Projection, TCA: True Component Analysis, PC: Principal Component, tSNE: t-distributed stochastic neighbor embedding.

adaptation or maladaptation following organ ischemia or pressure overload are both poorly understood and investigated.

In this study, we employ spatially resolved Raman microspectroscopy to obtain untargeted spatiomolecular information, exemplified using diseased cardiac tissues. While current approaches aim at genetics-based clustering, we used spatially-aware spectral information (i.e., integration of the spatial location of each spectrum) to generate an unsupervised clustering of tissue and cells, allowing us to identify matrical components of similar composition or cells of same cell type. We extended our analyses by investigating intra- and inter-cluster heterogeneity and performing spatial trajectory analyses. Surprisingly, the Raman spectromics approach was able to identify myocardial subclusters when approaching cardiac pathologies, which could not be disclosed by conventional (immuno-) histology. Furthermore, we combined Raman spectromics with spatial metabolomics to a multi-omics analysis of infarcted tissue, and extended Raman spectromics by multiplexed immunofluorescence to explore the immune cell landscape of myocardial infarction. To the best of our knowledge, this is the first attempt to in-depth, spatially-aware systems-level analyses based on Raman scan samples. Our method provides the unique opportunity to decipher both the cellular and subcellular biomolecular architecture of tissues and individual cells using a marker-independent, non-destructive approach based on molecular structures, rather than on cell markers or gene expression profiles.

## Results

### Study design and data analysis workflow

To establish our Raman "spectromics" approach we employed two widely used and robust mouse models in cardiovascular research: a) acute injury: myocardial ischemia/reperfusion injury (MI) by transient ligation of the left anterior descending artery (LAD) as described previously[16] and b) chronic injury: continuous infusion of Angiotensin II in atherosclerosis-prone apolipoprotein E (*ApoE*)-deficient mice, leading to cardiac hypertrophy and fibrosis[17] (Methods). The model of cardiac hypertrophy was selected to examine myocardial remodeling during chronic damage, while the MI model was suitable to explore infiltratory cells inside the infarcted heart and its inflammatory micromilieu.

The workflow starts with scanning deparaffinized formalin-fixed, paraffin-embedded (FFPE) or cryosections with a confocal Raman microspectroscope (Fig. 1a). Scans were acquired at 1 μm/pixel resolution, with each pixel being represented by an individual Raman spectrum, composed by wavenumbers of Raman shift and corresponding intensities. The resulting data matrix with more than 43,000,000 data points was filtered and subsetted subsequently (Fig. 1b). We analyzed the "whole spectrum" with wavenumbers from 300 to 3000 cm$^{-1}$ and compared it to the biological "fingerprint spectrum" between 400 and 1800 cm$^{-1}$, as defined previously[18]. Technical outliers and paraffin peaks as residues in FFPE tissue sections were filtered out, as detailed in the Methods section.

### Spatially-unaware Raman spectra show distinct clustering

Originally intended as exploratory analysis, we applied pipelines used for single-cell analyses to Raman spectroscopy datasets. While single-cell transcriptomic data provide information about genes and their transcript counts, Raman spectroscopy supplies data about wavenumbers and their corresponding intensities. Specific wavenumbers are a result of energy differences between incident and scattered photons and unique for individual molecules. Thus, wavenumbers can be interpreted like a gene transcript, but defining the actual material texture and molecular composition rather than genomic activity.

To test the hypothesis, we used Raman spectra acquired from an area of subendocardial fibrosis within a section from the hypertrophy model (Fig. 1d) and analyzed the dataset with the single-cell analysis

tool Seurat[19,20]. Outlier-corrected whole spectrum data was integrated into the Seurat workflow (Supplementary Fig. 1b) to generate Uniform Manifold Approximation and Projection (UMAP) plots (Fig. 1c). We observed a distinct segregation of pixels into clusters, as we expect similar spectra of the same cardiac substructure to be clustered together. The segregation also displayed a dependence on the number of dimensions used. In contrast to transcriptomics, Raman spectra variances between clusters are characterized by a high number of small changes. However, using more than 20 Principal components (PCs) did not further improve clustering, as Seurat's JackStraw- and ElbowPlot Analysis (Supplementary Fig. 3a–c) underlined.

The cluster analysis was carried out for both fingerprint spectrum (Fig. 1f) and whole spectrum (Supplementary Fig. 2). Interestingly, there was no difference in observed cluster numbers and an overall comparable clustering result. Consequently, the more precise fingerprint spectra were applied for further analyses. Figure 1g shows the average Raman spectrum for each identified cluster. We concluded that repurposing of established tools for genetic analysis to Raman spectral data is applicable and feasible.

### Cluster characteristics, spatial decoding and biological assignment

We sought to investigate whether the detected cluster can be translated to cardiac structures or compartments. For this purpose, we performed differential "expression" analysis (Fig. 2a) which can be repurposed to identify typical peaks in Raman spectra within a cluster. Clusters 0 and 1 were enriched in assignments to 1005, 1457 and 1640–1668 cm$^{-1}$ typically found in myocardium[10,11,21]. Cluster 2 was identified as area where no tissue covered the glass microscope slide. This cluster showed very low intensities overall with an enhancement between 400–600 cm$^{-1}$ and 1000–1200 cm$^{-1}$, resulting from background signal of glass. Cluster 3 demonstrated comparable peak intensities to cluster 4, but at a lower overall intensity and differences in relative peak ratios. Cluster 4 can be identified as collagen signature, showing characteristic peaks at 858, 940, 1249 and 1680 cm$^{-1}$ [22]. Cluster 5 was characterized by peaks at 889, 1056, 1302 and 1416 cm$^{-1}$ potentially resulting from spectral artefacts obtained by the removal of paraffin residues during preprocessing. Most relevant peaks expressed by cluster 6 were found at wavenumbers of 784, 1096, 1376 and 1580 cm$^{-1}$ often reported for nucleic acids[23,24]. Cluster 7 demonstrated the overall strongest peak intensities, especially in the region between 1200 and 1700 cm$^{-1}$ with peaks at 1123, 1343, 1401, 1605, and 1637 cm$^{-1}$. An overview of the most relevant peaks and their molecular assignment is provided in Supplementary Table S1.

Analysis of the spectral assignments of the clusters retrieved by the established unsupervised transcriptomics workflow enabled to identify biologically relevant spectral signatures of myocardium, collagens and cells and was further capable of separating them from preprocessing artefacts and background signals. Furthermore, the relative contribution of each component to the overall composition of the tissue could be determined (Fig. 2b). Overlaying density estimate contour lines on the UMAP plot (Fig. 2c) showed that cluster 0 and 1 to transition continuously into each other, while cluster 2, 6, and 7 showed greater compactness and hence homogeneity within the clusters found.

Next, we aimed to explore how specific peaks are distributed over the UMAP plot and specifically how these peaks spatially characterize the actual scanned sample. Cluster-specific peaks from the average spectrum and the differential "expression" analysis performed before were selected and the corresponding intensities were plotted in a color-coded fashion on a UMAP plot containing pixels sorted by UMAP projection, and on the actual image, where these pixels are sorted in their correct spatial relationship (Fig. 2d). Peaks for univariate intensity-based images were chosen exemplarily at 1457, 480, 858, and 1605 cm$^{-1}$ for clusters 0, 2, 4, and 7 to localize myocardium, glass,

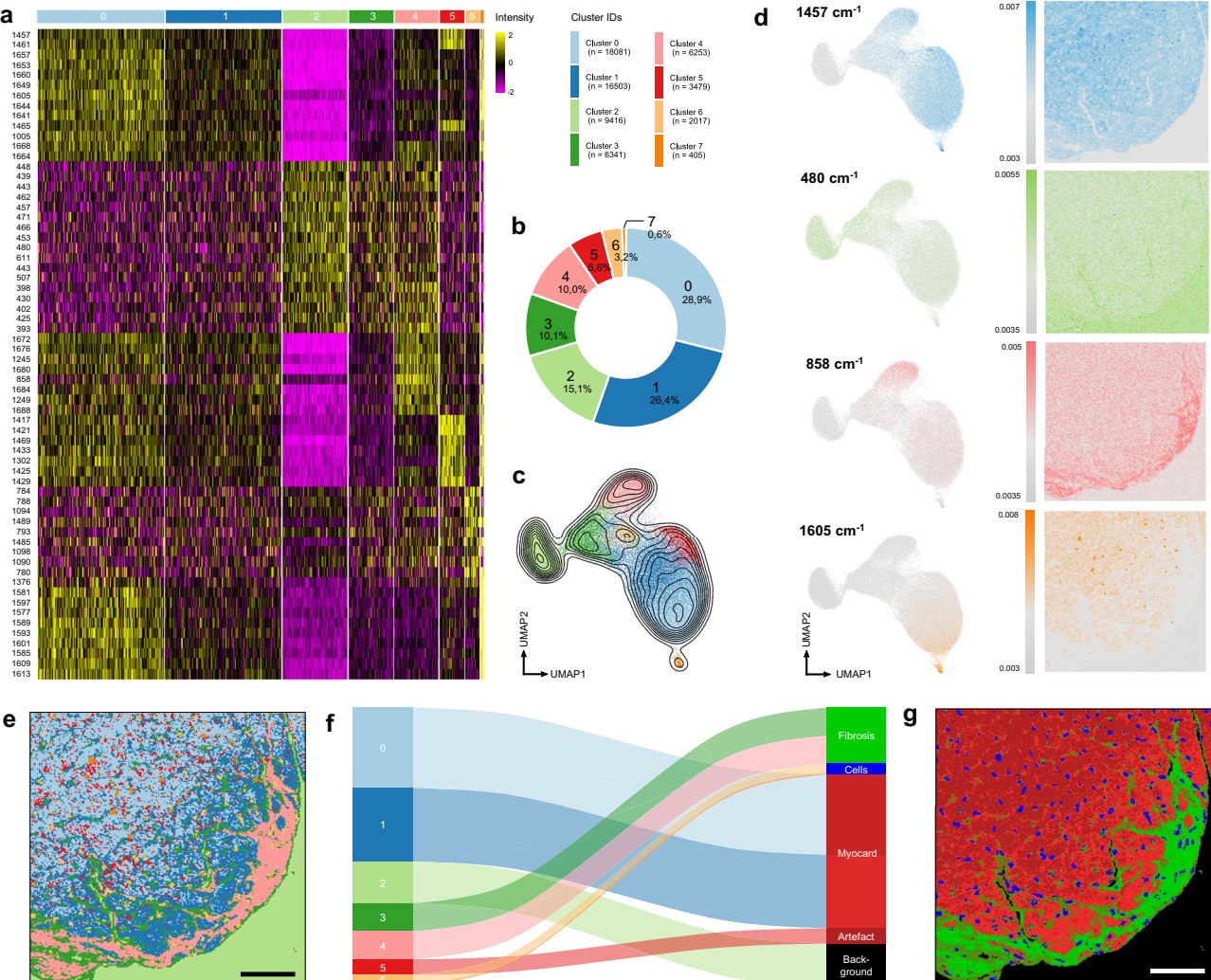

**Fig. 2 | Cluster characteristics, spatial decoding and biological assignment.**
**a** Heatmap of top differential Raman peaks per cluster, sorted by average log2 foldchange. Wavenumbers are rounded by one decimal place. **b** Cluster characteristics, n is number of pixels assigned to specific cluster, percentages are overall cluster size. **c** Density contour lines overlayed on UMAP plot, underlining compactness and hence similarity/homogeneity within clusters. **d** Intensity distribution of selected peaks over UMAP of analyzed pixels. Corresponding intensity distribution images with spatially reordered pixels. Key wavenumbers for cluster 0, 2, 4 and 7 (from top to bottom) were identified by DEG analysis and cluster-specific average spectra. **e** Pixels reordered in correct spatial correlation and colored by clusters identified from analysis before. **f** Sankey plot visualizing supervised assignment of detected clusters to myocardial substructures. **g** Supervised "ground truth" coloring of identified clusters for comparison with classical histological staining. Scale bar 50 μm. DEG: differential expressed genes.

collagens and blood-derived cells based on the spatial distribution along with molecular assignments to amide II and C-C backbone vibrations of proteins, characteristic glass peaks, hydroxyproline and hemin. The 858 cm$^{-1}$ Raman shift allowed for a distinct fit to collagen assignments, when comparing the intensity image (Fig. 2d) to the Picrosirius Red Staining of an adjacent section (Fig. 1d, right). Similar effects were found for the peak at 1605 cm$^{-1}$, which demonstrated nuclear morphologies in the intensity images and could be assigned to blood cells. Interestingly, peak assignments to 1605 cm$^{-1}$ were predominantly found in viable myocardium and not scar tissue. Whether different cell types can also be identified based on different spectra will be investigated later in this study.

To get back to the spatial dimension of the Raman scan, each pixel of the original Raman image was color-coded in its corresponding cluster color (Fig. 2e). The image showed clear spatial compartmentalization and strong similarities to the Hematoxylin and Eosin (H&E) and Picrosirius Red Staining (Fig. 1c), as well as the manually assigned Raman true component analysis (TCA) using commercially available software (Fig. 1e, Methods). Figure 2f illustrates the transition from the

unsupervised clustering to supervised ground truth cluster assignment of the Raman image by a combination of the molecular interpretation of the retrieved spectral clusters as well as morphological/spatial assignments. Strikingly, more than one cluster was found that could be assigned to distinct cardiac compartments. Two clusters were observed in the fibrotic tissue area, from which cluster 3 seems to surround cluster 4. Cluster 4 corresponds to the fibrotic area correlating well with the fibrosis staining, whereas cluster 3 was only detected by spectroscopic imaging and suggests representing a pre-fibrotic boundary around the actual fibrosis. The unsupervised analysis method also found two clusters allocated to myocardium: Interestingly, cluster 0 and 1 displayed a clear spatial organization reaching from far (cluster 0) to close (cluster 1) proximity to the fibrotic area. We hypothesized that this spatial pattern is a result of cardiac remodeling and hypertrophy when approaching subendocardial fibrosis. Cluster 1 could represent a transition zone between healthy myocardium (cluster 0) and fibrosis (cluster 4). As these results were found using unbiased, spatially-unaware clustering of Raman data, we concluded that our "Raman spectromics" approach could represent a

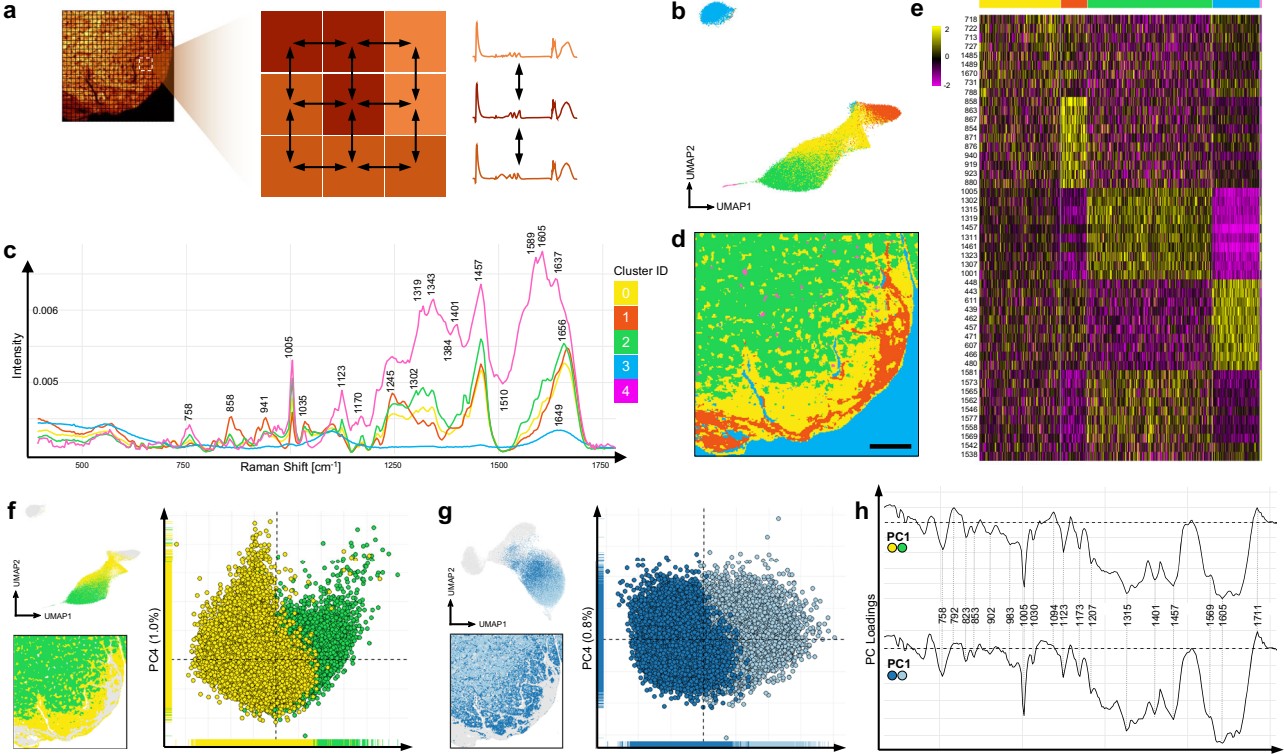

**Fig. 3 | Integration of spatial information confirms tissue clustering and myocardial zonations. a** Implementation of spatial context into Raman spectroscopy analysis using the bioinformatic model of a Markov random field (see Methods), which considers similar tissue to be closer together in space. **b**, **c** Cluster averages and UMAP projection identified by spatially-aware cluster analysis of fingerprint spectrum using BayesSpace. **d** Spatially reordered pixels, colored by cluster identified by BayesSpace. Scale bar 50 μm. **e** Heatmap of top differential intensities of Raman peaks per cluster, sorted by average log2 foldchange. **f, g** PCA to decipher molecular differences between healthy und remodeling myocardium, identified by spatially-aware (**f**) and spatially-unaware (**g**) cluster analysis. **h** PC loadings indicating changes in protein secondary structure towards ß-sheet structures and metabolic changes driven by myocardial remodeling.

sensitive method for identification of tissue substructures and hidden molecular patterns.

## Integration of spatial information confirms tissue clustering and myocardial zonations

To this point analysis of Raman spectra has been performed on pixels detached from their spatial context. By integrating spatial information about the position of the pixel inside the Raman scan image, we sought to validate the spatial-unaware clustering results shown before, as well as to reduce clustering artifacts which occurred before. As spatial transcriptomics have demonstrated previously, the additional spatial information can help address the analytical challenges of sparsity and noise by smoothing over adjacent pixels[25], which are more likely to have a similar genetic – or in our case Raman – fingerprint.

To analyze the dataset with spatial resolution, we translated a bioinformatics tool for spatial transcriptomic analysis into the "Raman world". BayesSpace provides a clustering method that uses a t-distributed error model to identify spatial clusters[25]. It implements a fully Bayesian model with a Markov random field (Fig. 3a), which hypothesizes that pixels belonging to the same type of cell or tissue should be closer to each other. We integrated Raman spectroscopy data into the BayesSpace workflow (Supplementary Fig. 1b) to cluster the previous sample of subendocardial fibrosis into 5 clusters for the fingerprint spectrum, or 4 clusters using the whole spectrum dataset (Supplementary Fig. 5). UMAP analysis of spatially-aware Raman data showed a completely different projection than without spatial resolution, with one cluster segregating far away from the others (Fig. 3b). This cluster could be easily identified as scan area without biological tissue. Notably, limiting the analysis from whole spectrum to the fingerprint area substantially improved clustering specificity and thus

allocation of myocardial structures (Supplementary Fig. 5). In summary, integrating spatial information of pixels containing spectral information allowed us to reproduce cluster analysis performed by spatially-unaware algorithms from above. On one hand spatially-aware clustering can reduce clustering artifacts by smoothing over adjacent pixels, but on the other hand, it could decrease cluster sensitivity and increases inclusion of outliers, which were "sorted out" by spatially-unaware clustering through assignment to a dedicated cluster. However, when benchmarking the accuracy for detection of fibrotic spectra, spatially-aware clustering using BayesSpace outperformed spatially-unaware clustering (Supplementary Fig. 6). Further analyses indicate that especially in the larger space the benefits of spatially-aware clustering come into play, e.g. in reducing artificial clustering (Supplementary Fig. 7).

Strikingly, performing spatially-aware clustering reproduced our previous finding of an additional myocardial subcluster between healthy myocardium and fibrotic subendocardium. We hypothesized that this cluster represents a transition zone of myocardium undergoing heavy remodeling, implying degradation and synthesis of cellular and interstitial components. To identify the underlying differences between healthy and remodeling myocardium, we performed Principal Component Analysis (PCA) between the two myocardial subclusters identified using both spatially-aware clustering via BayesSpace (Fig. 3f) and spatially-unaware clustering (Fig. 3g) via Seurat. However, the clustering by PCA was not as pronounced as by UMAP plotting. A shift of the myocardial remodeling cluster towards negative PC1 values was demonstrated for both input data (Fig. 3h). Major Raman peaks identified by the PC1 loadings were shown at 758, 823, 853, 1005 and in the amide I region between 1605–1650 cm⁻¹, suggesting differences in single amino acids such as tryptophane

(758 cm$^{-1}$), tyrosine (823 and 853 cm$^{-1}$) and phenylalanine (1005 cm$^{-1}$). The differences found in single amino acids as well as glycogen (853, 1022−1025 cm$^{-1}$) assignments imply differences in protein composition and metabolism during cardiac remodeling. Moreover, alterations in the shape of the Amide I band towards broader shoulders in the region between 1605−1650 cm$^{-1}$ indicate changes in protein secondary structure towards ß-sheet structures[26–28] potentially driven by myocardial remodeling.

To challenge the robustness of our untargeted spatial Raman spectromics approach, we reproduced clustering of healthy and remodeling myocardium on further sections of cardiac hypertrophy (Supplementary Fig. 8). We conclude that both spatially-unaware and -aware clustering algorithms applied on Raman spectroscopy data allows for the detection of compartments and zonations in histological sections at subcellular resolution.

### Deciphering intra- and intercluster heterogeneity by repurposing pseudotime and spatial trajectories

As shown before, unsupervised clustering algorithms could find myocardial zonations, which classical staining could not uncover. However, strict borders in the same tissue of myocardium must be critically questioned, as changes in the sense of remodeling tend to be of rather continuous nature. Hence, clustering into groups reduces important biological information by neglecting heterogeneity within a cluster. We aimed to visualize the transition from healthy to remodeling myocardium by deciphering intra- and intercluster heterogeneity within the two myocardial zonations found by spatially-unaware and spatially-aware cluster analysis.

To this end, we repurposed the bioinformatics tool Monocle[29,30], which performs pseudotime analysis on single-cell data. Pseudotime is a measure of how much progress an individual cell has made through a process such as cell differentiation, activation or throughout life. These cellular processes underlie changes in gene expression, leading to differential expression despite being of same cell type[29]. By translating this method to Raman microspectroscopy, we aimed to uncover tissue heterogeneity and homogeneity as wells as spatial correspondence of pseudotime trajectories. Instead of ordering cells with gene expression changes along a time axis, we sought to order pixels with Raman spectra along an axis and let this ordering be driven by changes in peak intensities of specific wavenumbers. We expected pixels with similar spectra to be close together and a continuous spectral change when moving along the pseudotime track. When there are too many different intensity changes that do not allow ordering on the main axis, a branch is constructed and spectra are placed here. Hence, branching is a result of strong spectral heterogeneity.

Monocle introduced an algorithm to learn the sequence of gene expression changes each cell must go through as part of a dynamic biological process[30]. In our case, we analyzed changes in Raman spectra instead of gene expression. Fingerprint spectrum data of the subendocardial fibrosis section used before was integrated into the Monocle environment (Supplementary Fig. 1b). Looking at the UMAP projection with overlayed pseudotime trajectory, we discovered a highly branched track in the region of myocardial clusters (Fig. 4a). This result is suggestive for a high heterogeneity within the myocardial spectra of our Raman analysis. To further elaborate this finding, we picked a region in the center of the Raman image where pixels were approximately equally assigned to healthy (cluster 0, light blue) or remodeling (cluster 1, dark blue) myocardium. These spectra were subsequently analyzed using the DDRTree method provided with Monocle (see Methods), which ordered the selected pixels/spectra along a track with several color-coded branches and subclusters (Fig. 4b). Unexpectedly, the algorithm found the branches to be located nearly exclusively within the assigned pixels of remodeling myocardium (Fig. 4c). This result underlines the dynamics of

remodeling of myocardium, while healthy myocardium is not affected by these changes.

We concluded that molecular dynamics are the result of vigorous degradation and synthesis of cardiac matrix during remodeling and can be illustrated by an unsupervised approach of translating pseudotime trajectories.

### Spatial trajectories towards fibrosis uncover molecular dynamics

To fully reveal the spatial organization of the tissue dynamics found, we employed spatial trajectories approximating the fibrotic regions, in order to examine the underlying molecular pattern changes. Linear trajectories towards subendocardial fibrosis were generated and the log2-normalized intensities of the fingerprint spectrum along this trajectory were plotted, with focus on pixels assigned to cluster 0 (healthy myocardium), 1 (remodeling myocardium), 3 (pre-fibrotic boundary) and 4 (fibrosis) (Fig. 4d) or just cluster 0 and 1 (Fig. 4i). Exemplarily, intensity shifts for hydroxyproline from collagen (858 cm$^{-1}$) and cytochrome c in cardiomyocytes (1318 cm$^{-1}$) were plotted in Fig. 4e, f. The increase of the collagen band indicates a pre-fibrotic remodeling of myocardium close to the region of subendocardial fibrosis und confirms our previous finding of a pre-fibrotic boundary around the actual fibrosis area. Excitingly, the alteration of cytochrome c intensity reflects the passage through healthy myocardium, remodeling zone and fibrotic area with alterations in mitochondrial metabolism at each level. The findings of a distinct myocardial subcluster of remodeling myocardium, as well as metabolic alterations approaching the fibrotic regions were reproduced in $n = 4$ individual mice with cardiac fibrosis (Fig. 4g, h). We validated our findings by exploring molecular dynamics along further trajectories, provided in the supplementary information (Supplementary Fig. 11).

The pseudotime analysis of the two myocardial subclusters in Fig. 4c demonstrated high molecular alterations in the remodeling subcluster (cluster 1) in contrast to that attributed to the healthy myocardium (cluster 0). To translate these findings to the spatial context, we analyzed intra-cluster heterogeneity along the trajectory, by filtering all pixels assigned to cluster 0 and 1 (Fig. 4i). Remarkably, we found high dynamics (i.e., intensity incline and/or decline) especially within cluster 1 of remodeling myocardium but not cluster 0 (Fig. 4j). These results support our previous findings of strong branching of the pseudotime trajectory, which was observed mainly within cluster 1. Peaks that showed strong dynamics along the spatial trajectory in cluster 1 were identified at 722, 1339, and 1569 cm$^{-1}$ and correlate to different vibrational modes of proteins such as the O-C-N bending (722 cm$^{-1}$), CN stretching and NH bending (1569 cm$^{-1}$) as well as C-C stretching of the protein backbone (1339 cm$^{-1}$) especially found in α-helical protein structures (Fig. 4j and Supplementary Fig. 11b)[31]. The latter finding was consistent with our previous finding of changes in protein secondary structure towards ß-sheets in remodeling myocardium. To quantify the spectral dynamics along all wavenumbers and also individual samples, the maximum variability of the local polynomial regression fitting (loess) curve and the derivation from the loess curve were calculated (see Methods). Especially quantification of the "curviness" of the intensity shift along the trajectory using the derivation from the loess curve showed significant differences within one sample ($p < 2.2e-16$, Fig. 4k, right) and also as reproduced in 4 individual mice ($p = 0.0458$, Fig. 4l, right).

### High-dimensional characterization of metabolic alterations in myocardial infarction by spatial trajectories and multimodal Raman-MALDI imaging

To challenge our findings derived from remodeling myocardium, we transferred our strategy of spatial trajectories to a murine MI model with transitory ischemia and consecutive reperfusion for 24 h. This results in an infarct area, surrounded by a hypoperfused area at risk

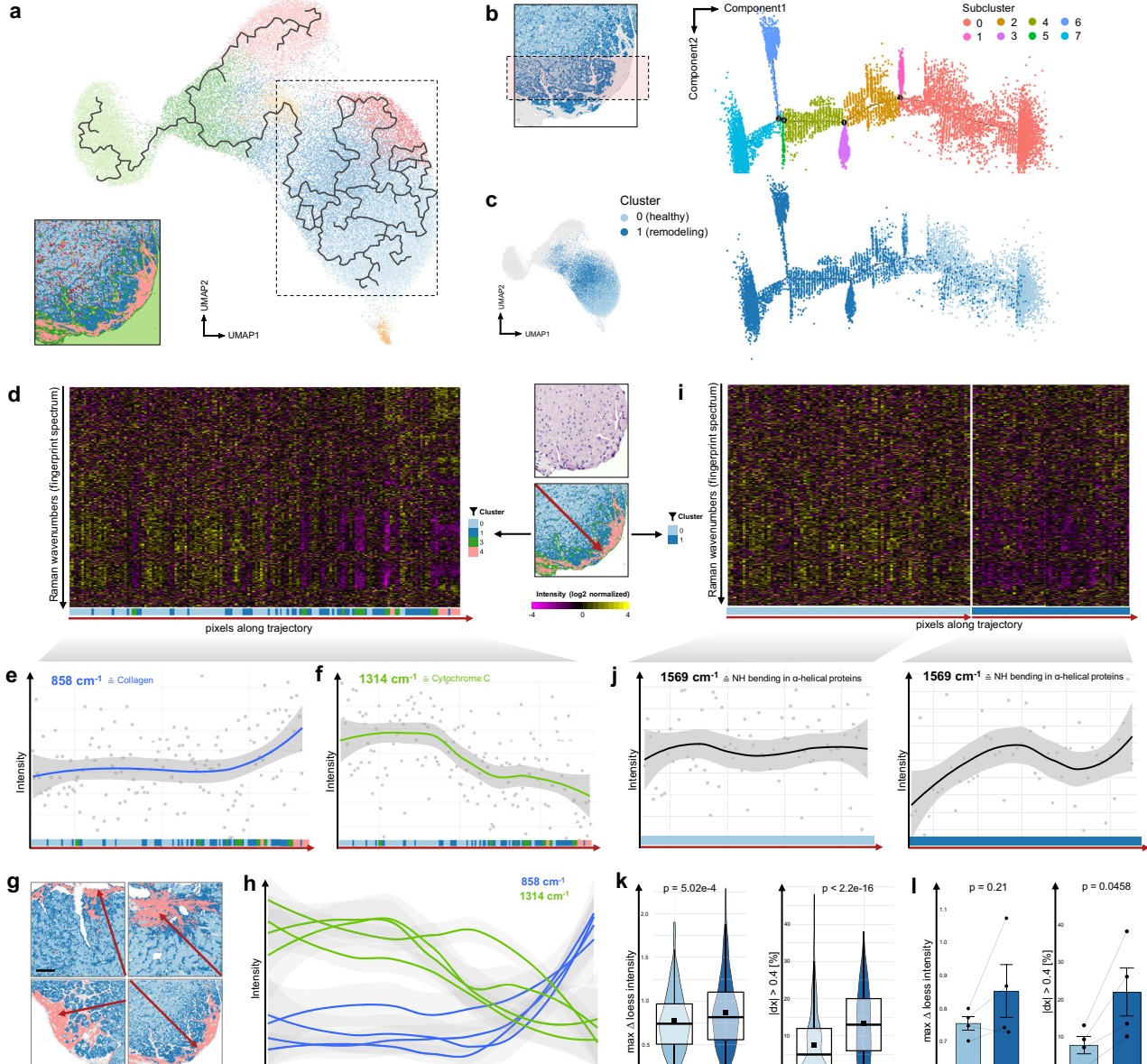

**Fig. 4 | Deciphering intra- and intercluster heterogeneity by repurposing pseudotime trajectories and employing spatial trajectories towards fibrosis.** **a** UMAP projection of spatially-unaware of Raman data with overlayed pseudotime trajectories. Branches denote crucial differences from pixels along the main trajectory and are predominantly found in clusters assigned to myocardium (dashed box). **b** DDRTree based pseudotime trajectory of pixels from center of Raman scan (area highlighted in red). **c** Branches almost exclusively occur with pixels assigned to cluster 1 (remodeling myocardium, dark blue), while spectra derived from healthy myocardium (light blue) appear homogenously. **d** Spatial trajectory along red arrow in section of subendocardial fibrosis. Heatmap of log2 normalized intensity changes in Raman fingerprint spectrum along this trajectory. Corresponding clusters along trajectory are plotted on bottom by their color code. **e, f** Selected wavenumbers with intensity shifts along the spatial trajectory. Blue lines are mean, grey ribbons are 0.95 confidence intervals. 858 cm$^{-1}$ band corresponds to hydroxyproline from collagen, 1314 cm$^{-1}$ to cytochrome c. **g** A similar myocardial pattern with a distinct healthy (light blue) and remodeling (dark blue)

cluster when approaching fibrotic regions (pink) was reproduced in 4 individual hearts. Scale bar 50 μm. **h** Reproduction of intensity shifts of collagen and cytochrome c ($n = 4$). **i** Log2 normalized intensity changes filtered to pixels assigned to cluster 0 and 1 along the spatial trajectory (red arrow). **j** Representative wavenumber demonstrating intensity dynamics within cluster 0 and 1. 1569 cm$^{-1}$ corresponds to NH bending especially found in in α-helical protein structures. Grey ribbons are 0.95 confidence intervals. **k** Violin plots showing significantly lower dynamics in all spectra from cluster 0 (light blue, healthy myocardium) in comparison to cluster 1 (dark blue), implying molecular dynamics of myocardium under remodeling. Quantification by calculating the maximum variability of the local polynomial regression fitting (loess) curve (left, $p = 5.02 \times 10^{-4}$) and thresholding of the derivation from loess curve (right, $p < 2.2 \times 10^{-16}$). Welch two-sided, paired t-test. $n = 1$ (sample from a). **l** Reproduction of spectral dynamics in the remodeling subcluster in $n = 4$ individual hearts. Welch two-sided, paired t-test. Bars display mean ± SEM.

with heavy cell infiltration, transitioning into healthy myocardium which was not affected by the coronary occlusion[32] (Fig. 5a). We sought to spatially reconstruct molecular changes when approaching the infarcted area, with focus on metabolic alterations. As results of ischemia, myocardium behind the ligated coronary artery undergoes a

rearrangement of energy metabolism, which is – among others – characterized by a shift of glucose metabolism from oxidative phosphorylation to enhanced glycolysis[33].

We created a spatial trajectory targeting the ligation site, which could be clearly identified in the H&E staining (Fig. 5a). Intensity

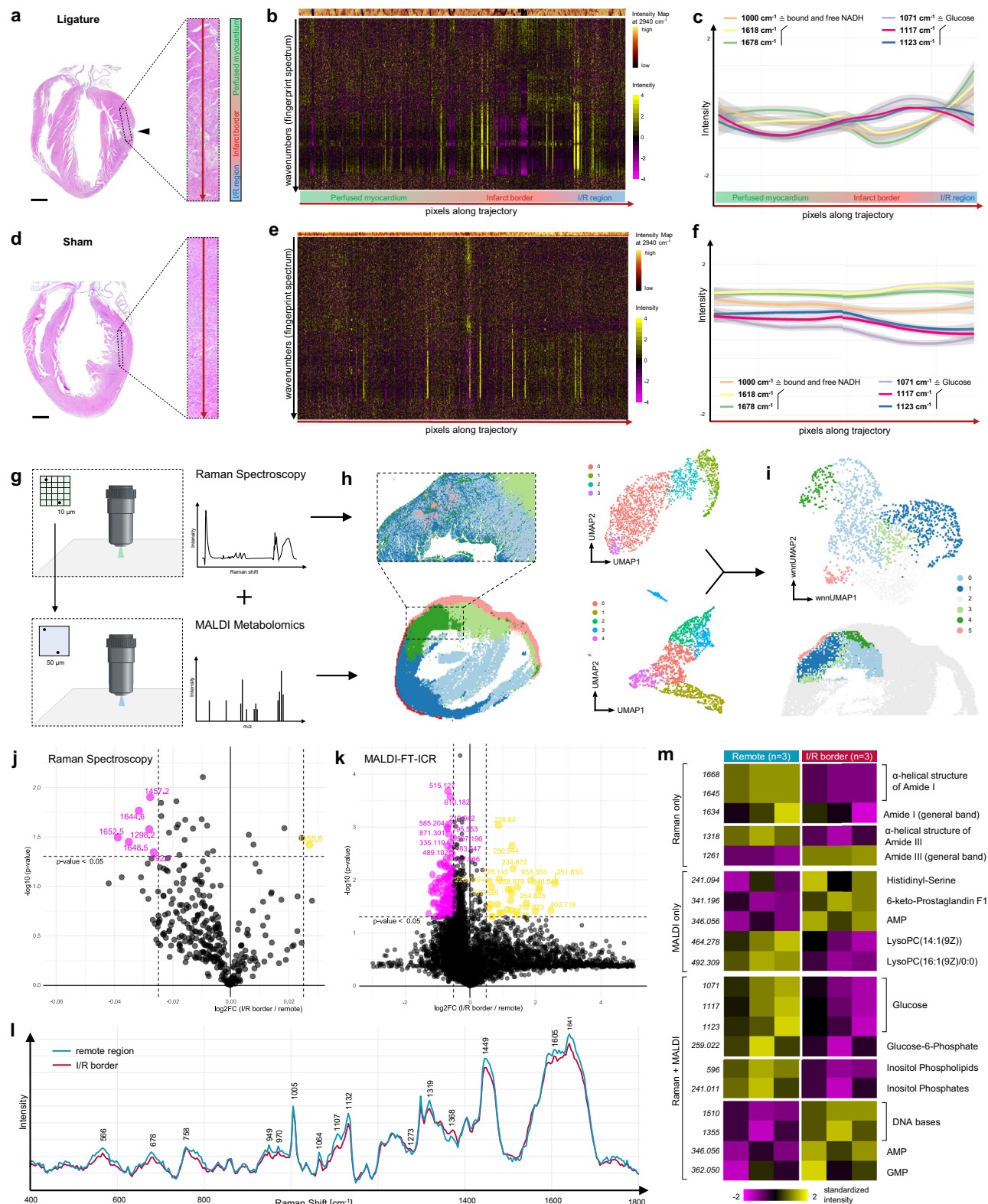

changes of the fingerprint spectrum along this trajectory were plotted in a heatmap (Fig. 5b). We additionally supplied the analysis with an intensity map of the scanned area for C-H vibrations in proteins (2940 cm$^{-1}$) to verify there is no general intensity shift along the large scan area. Employing spatial trajectories, we were able to spatially reconstruct metabolic changes from healthy myocardium towards the ischemic region. Three characteristic peaks for NADH[34,35] and glucose[34,36] were tracked along the trajectory and a specific pattern

when crossing the infarct border and entering the ischemic heart region (Fig. 5c) was observed: NADH was decreased at the infarct border and elevated in the ischemic region, suggesting a mitochondrial compensatory mechanism for energy delivery in the I/R region and marked tissue damage at the infarct border. In contrast, glucose was increased in the peri-infarct area, but decreased in adjacent areas. We hypothesized that glucose is shifted to the inflammatory site of the myocardial infarction, where inflammatory cells massively infiltrate

**Fig. 5 | High-dimensional characterization of metabolic alterations in myocardial infarction by spatial trajectories and multimodal Raman-MALDI imaging. a** H&E Staining of a murine sagittal heart section after induction of myocardial infarction by transient ligature of the LAD. Dashed box denotes cardiac tissue with transition from healthy to ischemic/reperfused (I/R) myocardium. Black arrow points at the ligation site. Long red arrow depicts spatial trajectory passing the infarct border and heading towards the ischemic heart region. Scale bar 1 mm. $n = 1$ individual sample. **b** Heatmap of log2 normalized intensity changes in Raman fingerprint spectrum along this trajectory. **c** Intensity shifts of three characteristic peaks for NADH and glucose were tracked along the trajectory and show a specific pattern when crossing the infarct border and entering the ischemic heart region. Grey ribbons are 0.95 confidence intervals. **d**–**f** Intensity alterations in a sham-operated heart, where the suture around the LAD was not closed. Grey ribbons are 0.95 confidence intervals. $n = 1$ individual sample. Scale bar 1 mm. **g** Illustration of the multi-omics approach of combining Raman and MALDI imaging on the same tissue regions. For best fitting, resolution of Raman imaging was 5 times higher than MALDI imaging (10 vs. 50 μm). **h**, **left** Cluster analysis of both datasets separately. Both methods uncover distinct spatially organized clusters at the infarct border region. **h**, **right** Cluster analysis of overlaying data points after spatial co-registration of Raman and MALDI scans. **i** Integration of these datasets into a multimodal analysis using weighted nearest neighbor (wnn) dimension reduction and cluster analysis (top). The resulting cluster image (bottom) fits well with the boundaries of the infarcted region identified in the H&E staining (Supplementary Fig. 12). **j**, **k** Volcano plot of $n = 3$ individual hearts exploring spectral differences from cluster-identified I/R border regions vs. remote (healthy) regions. Welch two-sided, paired t-test. **l** Average Raman spectrum for I/R border regions and remote regions. **m** Heatmap of standardized mean difference between I/R and remote regions from $n = 3$ individual hearts. Raman-only section with modalities only identifiable by Raman spectroscopy, MALDI-only those using MALDI imaging. Raman + MALDI show reproducibility of measurements across the different methods. Raman wavenumbers (no decimals) and m/z values (three decimal places) are shown on the left of the heatmap.

the diseased cardiac tissue and consume large amount of energy, as well as increased glycolysis of cardiomyocytes. In contrast, sham-operated mice didn't show these metabolic alterations (Fig. 5d–f).

To further validate our findings and to supplement Raman spectroscopic data with more specific metabolic information, we combined our Raman "spectromics" approach with spatial metabolomics (Fig. 5g). Directly adjacent paraffin sections of 3 individual infarcted hearts were used to perform both Raman and MALDI imaging on the same region. Next, both datasets were integrated into the Seurat workflow and cluster analysis of the individual datasets was performed. Projecting the found clusters back to their spatial context showed for both Raman and MALDI datasets a distinct organization (Fig. 5h, left). Next, both datasets were spatially aligned (Methods), resulting in pixels containing both Raman and MALDI information. Dimension reduction and cluster analysis was then performed for both datasets individually and as multimodal analysis based on a weighted combination of the datasets. Strikingly, when translating the analyzed pixels back to their spatial context, we found a different spatial organization compared to those found by Raman or MALDI alone. The tissue organization identified by the multimodal analysis was to the best to reflect the ground truth boundaries of the infarcted region, compared to the H&E staining (Supplementary Fig. 12). Multimodal analysis on $n = 3$ individual infarcted hearts identified the clusters assigned to remote (healthy) and I/R (ischemia/reperfusion) border region and the spectral differences were visualized in volcano plots for both Raman (Fig. 5j) and MALDI (Fig. 5k) imaging. Both methods demonstrated an overall downregulation of metabolites in the infarct border region, as can be seen as a shift to the left in both plots. Significant and strong regulations of wavenumbers (Raman) or m/z values (MALDI) were plotted in a color-coded fashion (yellow: up, pink: down). We also plotted the average Raman spectrum from all hearts derived from the healthy remote region and the I/R border region (Fig. 5l).

Both Raman and MALDI imaging demonstrated to provide helpful information to define metabolic alterations in infarcted hearts. However, some of these information can technically or due to incomplete evidence only be identified by Raman or MALDI (Fig. 5m). As an example, Raman spectroscopy only identify a degradation of the α-helical structure of Amid I in proteins, together with an overall reduction of Amid I intensity in the ischemic region. For Amid III we found the same pattern for α-helical structures, but an intensity increase. Taken together with our previous results of changes in protein secondary structure towards ß-sheets in remodeling myocardium, this finding in the acute pathology provides further evidence on mechanisms of cardiac remodeling in the early stage of myocardial tissue damage.

Other metabolites could only be identified by MALDI analysis. Histidinyl-Serine as an example is a breakdown product of protein catabolism and cell signaling (HMDB0028894), which was increased at the infarct border. 6-Keto-Prostaglandin F1 is a marker of platelet activity (HMDB0002277) and was increased at the infarct region. The Lysophosphatidylcholines (LysoPCs) enlisted have anti-inflammatory effects (HMDB0010380, HMDB0010383) and were reduced in the infarcted area.

We also used the combination of Raman and MALDI to validate the identified metabolites by different methods. As an example, we compared 3 characteristic glucose peaks from Raman spectroscopy, as also used previously, and compared their intensity to that of Glucose-6-Phoshate identified by MALDI imaging. We found comparable results for Glucose, and also Inositol-Phosphate compounds and DNA bases.

Thus, sophisticated usage of Raman spectromics data can be used to spatially resolve metabolic changes, multimodal analysis as well as molecular alterations along clusters identified by unsupervised algorithms of multiple datasets.

## Defining the surrounding cellular (immune-) landscape in acute myocardial infarction

Following ischemia and reperfusion of the infarcted heart, blood-borne immune and inflammatory cells are a prominent feature in the diseased tissue. To explore the cellular environment of the infarcted heart, we transferred our strategy of Raman spectromics to the inflammation site of infarcted hearts. In order to define cellular sub-types by spectral data, the periinfarcted region was identified by H&E staining and consequently Raman scans from this region were overlayed by cyclic multicolor immunofluorescence staining (MACSima™ Imaging Cyclic Staining) of the identical specimen at the exact same position (Fig. 6a–d). This approach allowed direct identification of cell types by immunolabeling and allocating the underlying spectral fingerprint of the corresponding cell.

After selecting pixels with positive immunolabeling, dimension reduction of the underlying Raman spectra was performed and visualized using t-distributed stochastic neighbor embedding (tSNE) (Fig. 6f). Cardiomyocytes represented the largest cellular cohort of the analysis and presented the most heterogenous spectral composition. α-SMA+ Myofibroblasts or vascular smooth muscle cells (vSMCs) made up an own cluster, which did not clearly separate from the cardiomyocyte spectrum. In contrast, hemin+ erythrocytes and CD45+ leukocytes showed a clear clustering and separation from other cell types. CD41+ platelets ware mainly located within the immune cell cluster, however, this could also be the results of platelet-leukocyte-coaggregates. We also performed a subphenotyping of the immune cells (Fig. 6g): Ly6G+ neutrophils represented the largest and most heterogenous cluster, containing also Raman spectra from CD11b+ cells. MHC II+ professional antigen presenting cells (pAPCs) and CD68+ macrophages by contrast showed a specific clustering. We used

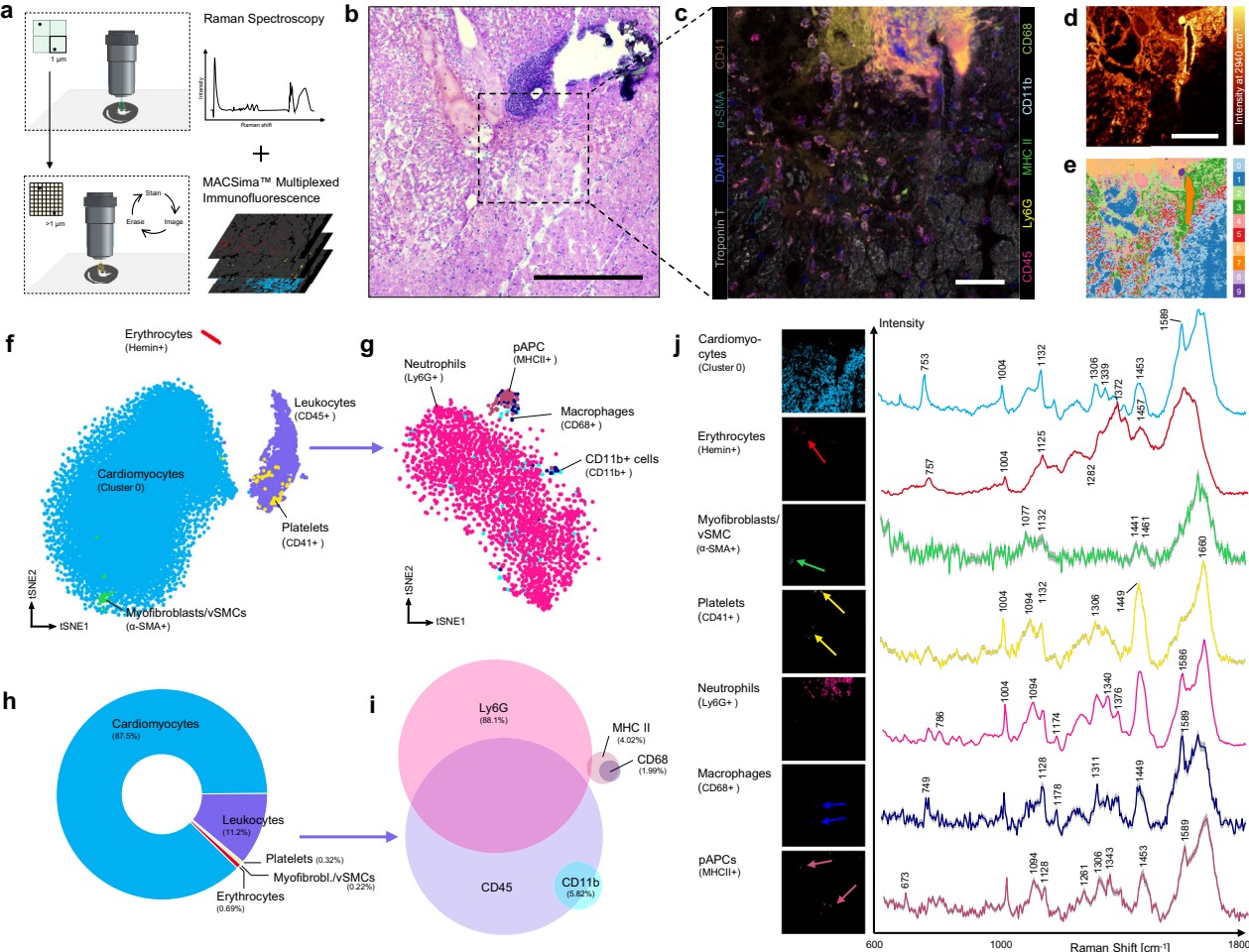

**Fig. 6 | Defining the surrounding cellular (immune-) landscape in the model of acute myocardial infarction. a** Multi-omics approach to decipher spectra from cells by combination of Raman spectroscopy with consecutive MACSima™ multi-color immunofluorescence staining. **b** H&E staining of an adjacent section of murine myocardial infarction. Scale bar 300 μm. **c** MACSima™ multiplexed immunofluorescence imaging was performed on the identical section and region as Raman spectroscopy was done previously. Scale bar 50 μm. n = 1 individual sample. **d** Raman Intensity images at 2940 cm⁻¹ (general band for lipids and proteins). Scale bar 100 μm. **e** Cluster image identified by the Seurat workflow. Cluster 0 was assigned to myocardium. **f** tSNE plot after dimensionality reduction of Raman spectra of identified cell types. Clear separation into cardiomyocytes, erythrocytes and immune cells. **g** Spectral subphenotyping of immune cells. Neutrophils display the largest cluster while MHC II+ professional antigen-presenting cells (pAPCs) and CD68+ macrophages separate into a distinct cluster. **h** Donut chart showing frequency distribution of identified cells (percent from absolute number of analyzed number of pixels). **i** Venn diagram of surface markers analyzed for the immune cell subpopulation. E.g., most Ly6G+ cells were also CD45 + . **j** Average spectra for the different cell types and characteristic peaks, together with the spatial representation of the analyzed pixels. vSMCs: vascular smooth muscle cells.

a donut chart and venn diagram to visualize the frequency distribution of detected cells and intersection of immune cell surface markers (Fig. 6h, i). Characteristic average spectra for each identified cell type are presented in Fig. 6j, together with the spatial location of each cell type. The same approach was performed with a FFPE section, showing comparable results in spectral signatures from the identified cell types (Supplementary Fig. 14). We further assessed accuracy on how well Raman spectroscopy can delineate individual cell types found in the disease myocardium. We used the cell type-specific average spectra found in our previous analysis as "reference spectra" and calculate the similarity between the reference spectrum and each pixel spectrum of the scan. The computationally identified cell types were then compared to the corresponding immunofluorescence staining as ground truth measure. Our analyzed data suggest that Raman spectroscopy has a high specificity but low sensitivity when detecting and delineating cell types (Supplementary Fig. 16). Except for erythrocytes, which have strong characteristic spectra and low heterogeneity, sensitivity was overall around 65%. Hence, Raman spectroscopy has a good potential to exclude the existence of specific cell types in a scan, but cannot securely define if these occur in the scan.

Thus, by this proof-of-concept analysis we could demonstrate two things: 1) Raman spectromics can be easily combined with further downstream analyses of the very same histological. 2) By manual selection of cell type-specific Raman spectra, a database of characteristic spectra can be derived which can in turn be used for future identification of cell types without need for additional immunofluorescence labeling and imaging. Although Raman spectroscopy provides high sensitivity, its efficacy in accurately detecting and distinguishing different cell types within a Raman scan remains an ongoing challenge.

## Discussion

In this manuscript, we showcase the potential of defining the spatial distribution of biomolecules by sophisticated translational analysis of data acquired by spatially-resolved Raman microspectroscopy. The approach offers a means to unravel homogeneity, heterogeneity, and dynamics within tissues and cells at a subcellular resolution, as shown in two models of cardiac pathologies.

In our quest to assess the effectiveness of our methodology, we conducted comparative evaluations against conventional Raman data

processing pipelines and deep learning models. Traditional techniques such as principal component analysis (PCA) and k-means clustering fall short in harnessing spatial information, thus neglecting valuable information. In contrast, the utilization of deep learning models – specifically employing latent features from autoencoder models – holds great promise for Raman spectral analysis[37,38]. We trained three distinct deep learning models, and their clustering outcomes were compared to that of our proposed approach. Here, deep learning was only partially able to detect and replicate biologically relevant clusters which our Raman spectromics clustering proficiently delineated (Supplementary Fig. 17).

Moreover, our approach evades the "black box problem" intrinsic to deep learning and obviates the necessity for a priori training of deep learning models. This positions our methodology as a more transparent and interpretable alternative to deep learning. However, the full advantage over deep learning models and other related approaches remains to be fully demonstrated, particularly in the context of large datasets. Deep learning's efficacy and potential emerges when a sufficient amount of high-quality labeled data is available, which in itself can entail substantial labor and time investment. Interpretability of deep learning models in the context of Raman spectra can be an additional hurdle. Unlike genomics or image data, where features can be directly associated with specific genes or structures, the features extracted from Raman spectra may lack clear chemical or molecular attributions, making it challenging to validate and explain the model's predictions.

The introduction of the term "Raman spectromics" underscores our intention to emphasize a comprehensive, holistic, and systems-level analysis of the acquired data. Integrated into a robust framework, multimodal analyses in combination with other spatial profiling techniques or translation to other bioinformatic analysis platforms are made possible. While other machine learning approaches lack interpretability, reproducibility and transparency, the presented methodology uses existing, well-established and well-recognized bioinformatics tools for transcriptomic analyses without "reinventing the wheel". Hence, the presented approach aligns with a strategy that prioritizes transparency, reproducibility, and interpretability. Nevertheless, the full potential, generalizability and applicability for biomedical research remains a subject of ongoing investigations.

Beyond the bioinformatic strides that our methodology presents, the method of Raman spectroscopy itself has the potential to characterize complex spatiomolecular information which cannot be achieved by other current 'omics technologies. Furthermore, Raman spectroscopy preserves both contextual and structural information of the specimen without necessitating tissue digestion or modification. On one hand, this brings the advantage of a limitless field of application, on the other hand, this allows combining our approach with further downstream analysis with the very same specimen, including staining, spatial transcriptomics or proteomics, or application in a living system, reaching from cell culture to human beings. Starting in basic science, Raman spectromics can be applied in preclinical and clinical research – including ex vivo and in vivo imaging. In addition, our proposed strategy might be applicable as diagnostic tool in sparse sample material and is compatible with both fresh frozen as well as paraffin-embedded tissues, opening the door for retrospective analysis of samples collected over decades in biobanks.

Another benefit of Raman microspectroscopy is the high resolution. As an example, our resolution of $1\,\mu m/px$ exceeds the current resolution of spatial transcriptomics nearly by the factor 7000, when comparing to 10X Genomics' Visium platform[39].

However, our methodology is not devoid of limitations. A key challenge is to address large data processing. The entire pipeline, starting from high-dimensional Raman data acquisition and extending through the analytical processes, necessitates careful consideration of the computational burden posed by voluminous data quantities.

Although the application of Seurat's standard workflows proved to be fast, the introduction of spatial resolution using BayesSpace's workflow notably increased computation time for the mentioned samples. Future advancements in big data management and spatial clustering algorithms will be indispensable to explore larger tissue areas. Additionally, certain limitations intrinsic to Raman spectroscopy itself, such as interpretational constraints due to either missing evidence or technical limitations, need to be addressed.

Looking forward, the unbiased exploration of spatially-resolved molecular patterns through Raman spectromics holds promise to provide insights in the fundamental principles that guide tissue remodeling and organization and will facilitate the study of the mechanistic basis of these patterns and their consequences. As the field continues to evolve, researchers could focus on developing hybrid approaches that combine the strengths of deep learning models with the interpretability and robustness of translational tools. Deeper biological insights gained this way, based on spatiomolecular tissue composition, will help to uncover the transition from physiological to pathological states and shed light on the spatial boundaries of disease.

## Methods

### Ethical statement

All animal procedures were performed according to the German animal protection law and approved by the local authorities (Regierungspräsidium Tübingen, TVA M5/17, M20/15, M04/19 G, M02/20 G and M01/21 G). Mice were housed under specific pathogen-free conditions at the University of Tuebingen. All experimental animals were kept in individually ventilated plastic cages in an air-conditioned and light-conrolled room. Mice were fed with a maintenance diet (Altromin 1324) ad libitum. To reduce the number of experimental animals, tissue samples were reused from experiments included within other projects. The data gained in this novel analysis method is genuine and therefore no data is published elsewhere.

### Animal experiments and sample preparation

**Myocardial infarction (MI) model.** To induce myocardial infarction, the left anterior descending artery (LAD) of 5 anaesthetized and ventilated 10–12 weeks old male mice (wildtype siblings of Pf4Cre Ackr3 knockout stain *B6.Cg-Thy1a-(Ackr3)Ackr3^{tm1Fma}-Tg(Pf4-icre)Q3Rsko/J*, bred in house) was transiently ligated as described before[16]. Briefly, mice were anaesthetized by a triple narcosis of midazolame (5 mg/kg body weight), medetomidine (0.5 mg/kg body weight), and fentanyl (0.05 mg/kg body weight). After lateral thoracotomy, the LAD was occluded with a placeholder. After 30 min of ischemia, the reperfusion period was sustained for 24 h. Mice were kept under buprenorphine analgesia (0.1 mg/kg body weight), which was administered every 8 h. 24 h after LAD ligature, mice were sacrificed by cervical dislocation under isoflurane anesthesia (5%v/v), and the heart was extracted for histological analysis.

**Hypertrophy model.** Osmotic mini-pumps (model 1004, Alzet, Durect Corporation, Cupertino, CA, USA) were subcutaneously implanted into 12 ApoE^{-/-} mice (*B6.129P2-Apoe^{tm1Unc}/J*, Charles River, Massachusetts, USA) at 8–12 weeks of age through a small incision in the back of the neck of anesthetized mice. The pumps delivered Angiotensin II (Cat. A9525, Sigma-Aldrich, St. Louis, Missouri, USA) at a continuous delivery rate of 1000 ng/min per kg body weight. Mice were sacrificed after 28 days of pressure-overload injury, resulting in cardiac hypertrophy and remodeling.

**Tissue preparation.** After sacrifice of mice, hearts were rinsed with PBS containing 20 U Heparin, explanted and fixed in 4% formalin for at least 24 h. Subsequently, hearts were dehydrated, embedded into paraffin and serial sections were cut at 5 μm using a microtome (Leica). Before

stainings and Raman scans, sections were deparaffinized by 3 changes in Xylene (Carl Roth, Karlsruhe, Germany) for 10 min and consequently sections were rehydrated. For cryosections, hearts were immediately placed in a cryomold with O.C.T. (Tissue-Tek®) after explantation. Subsequently, cryomolds were transferred into an aluminium pan filled with 2-methyl butane (Sigma) and dipped into liquid nitrogen. After complete freezing, hearts were stored at −80 °C until experiments.

**Standard histological stainings.** For hematoxylin and eosin stainings (H&E), deparaffinized and rehydrated sections were stained in hematoxylin (Sigma-Aldrich) for 5 min, washed for 15 min under running tap water, and consequently stained in eosin (Carl Roth) for 3 min.

Fibrosis staining was performed by Picro-Sirius Red staining, including incubation of deparaffinized and rehydrated sections with 1.2% Picro-Sirius Red Solution (Merck, Darmstadt, Germany) for 60 min and subsequent washing in 0.5% acetic acid.

## Image acquisition

**Microscopy for histological stainings.** Images of standard histological stainings were acquired using a Nikon Eclipse Ti-2A microscope with 4-40x objectives and Nikon Software NIS-Elements AR (Nikon, Tokyo, Japan).

**Raman microscopy.** For Raman analysis, FFPE sections were deparaffinized and rehydrated as described above. No staining or further tissue preparation was needed (Supplementary Fig. 1a). Raman imaging was performed on a Witec alpha 300 R confocal Raman microscope (Witec GmbH, Ulm, Germany). Sections were scanned with a green laser (532 nm) at 50 mW laser power, an integration time of 0.05 s/pixel and at 63x objective. The scan areas were mostly 250 × 250 pixels or 300 × 300 pixels at a resolution of 1 μm per pixel. Large area scans were performed at a resolution of 10 μm per pixel. For spatial trajectories and large area scans, a topography correction was performed by true surface manual learning of 5 × 5 surface points or 3-point-plane-correction.

**High mass-resolution MALDI-FT-ICR-MSI analysis.** The MALDI-FT-ICR-MSI analyses were performed as previously described[40–42]. In brief, FFPE heart samples were cut into 5 μm sections on a microtome (Leica) and mounted onto indium-tin-oxide coated conductive glass slides (Bruker Daltonik, Bremen, Germany). The FFPE sections were incubated at 60 °C for one hour and deparaffinized in xylene (twice for 8 min). The matrix solution consisted of 10 mg/ml 9-aminoacridine hydrochloride monohydrate (9-AA) (Sigma-Aldrich, Germany) in water/methanol 30:70 (v/v). SunCollect sprayer (Sunchrom, Friedrichsdorf, Germany) was used for matrix application. The flow rates were 10, 20, 30 and 40 μl/min, respectively, for the first four layers. The other 4 layers were performed at 40 μl/min. The MALDI-MSI measurement was performed on a 7 T Solarix XR FT-ICR mass spectrometer (Bruker Daltonik, Bremen, Germany) in negative ion mode using 50 laser shots per spot at a frequency of 1000 Hz. The MALDI-MSI data were acquired over a mass range of m/z 75–1000 Da with 50 μm lateral resolution. MALDI mass spectra were root mean square normalized using SCiLS Lab (Bruker Daltonik, Bremen, Germany), and exported as imzML files. MS peak annotation was performed with Human Metabolome Database[43] (http://www.hmdb.ca/) and METASPACE[44] (https://metaspace2020.eu/).

**Cyclic multicolor immunofluorescence staining using the MACSima™ platform.** After Raman spectroscopy scan, samples were stored in PBS buffer for up to 24 h. Heat-induced antigen retrieval was performed at basic conditions using a TEC buffer (2 mM Tris, 2 mM EDTA, 1 mM Sodium Citrate, pH 9.0) and boiling the samples for 20 min. Samples were cooled down and stored in MACSima™ Running

Buffer (Cat 130–121-565, Miltenyi Biotec, Bergisch Gladbach, Germany) until initial DAPI staining. After DAPI staining for 10 min in the dark, samples were washed three times with MACSima™ Running Buffer and subsequently submitted to MACSima™ Imaging Cyclic Staining (MICS). The MACSima™ device is an ultra-high content immunofluorescence microscope, performing fully automated cycles of fluorochrome-labelled antibody staining, image acquisition, and fluorochrome bleaching. Images were generated according to the manufacturer's instructions and analyzed with the MACS® iQ View image analysis software (Miltenyi). Used antibodies are enlisted in Supplementary Table S2.

## Data analysis

**Raw data preprocessing.** Images were acquired at 1 μm/pixel resolution, resulting in a matrix of 62,500 pixels for a scan area of 250 by 250 μm, or 90,000 pixels for 300 by 300 μm (for combination with immunofluorescence staining). This led to a total of more than 62,000,000 data points (90,000 columns x 692 rows). Raw spectral data was preprocessed by cosmic ray removal, background subtraction and baseline correction using Project FIVE 5.2 (Witec, Ulm, Germany). All methods for data preprocessing haven been described before[45]. We analyzed the "whole spectrum" with wavenumbers from 300 to 3000 cm$^{-1}$ and biological "fingerprint spectrum" between 400 and 1800 cm$^{-1}$, as defined previously[18]. Technical outliers, identified as extreme values of the spectra, were filtered out. A limitation in use of formalin-fixed, paraffin-embedded (FFPE) tissue sections is artifacts caused by paraffin residues leading to sharp, distinct peaks, which may lead to unwanted clustering of pixels. Hence, areas contaminated with paraffin were detected as distinct subcluster (Supplementary Fig. 2b). For reasons of reproducibility we decided to exclude the paraffin peaks occurring at wavenumbers of 1064, 1132, 1294 and 1441 cm$^{-1}$, which have been described in literature before[46]. For cryosections, no paraffin removal was applied.

**Raman true component analysis (TCA).** TCA is a method provided by Witec Project FIVE software (Witec) and has been described before[18,45]. Briefly, TCA identifies spectral components that are most prevalent in the dataset by a non-negative matrix factorization-based algorithm that defines similar spectra as the same component. These components are visualized by generating color-coded intensity distribution images.

**Spatially-unaware cluster analysis.** The R Package Seurat[47] was used for spatially-unaware cluster analysis. Seurat allows for the integration of multiple data types across different technologies to identify and interpret sources of heterogeneity from single-cell transcriptomic measurements[19,20]. After quality control of integrated data (outlier correction, paraffin peak removal), Raman spectroscopy data was integrated into the Seurat workflow (Supplementary Fig. 1b). For dimension reduction, principal component (PCA) analysis provided with Seurat was used. For cluster analysis, the number of dimensions according to Seurat's ElbowPlot and JackStrawPlot were utilized (Supplementary Fig. 3). Furthermore, impact of different cluster resolutions were investigated using the R package "clustree"[48]. T-distributed stochastic neighbor embedding (tSNE) was run with perplexity = 30.

**Spatially-aware cluster analysis.** The R package BayesSpace[25] was used for spatially-aware clustering. BayesSpace uses a t-distributed error model to identify spatial clusters. This model is particularly robust against outliers and technical noise in clusters, which may be driven by technical artifacts during sample preparation or image acquisition. BayesSpace implements a fully Bayesian model with a Markov random field, which hypothesizes that pixels belonging to the same type of cell or tissue should be closer to each other. Doing so,

BayesSpace substantially outperforms all current non-spatial clustering algorithms and spatial clustering methods developed for spatial transcriptomic data[25,49]. After quality control of the data (outlier correction, paraffin peak removal), Raman spectroscopy data was integrated into the BayesSpace workflow (Supplementary Fig. 1b). For cluster analysis, the number of clusters located at the "elbow" of the negative pseudo-log-likelihood curve (Supplementary Fig. 5a, e) was selected. UMAP analysis was performed by scater package[50]. Heatmap analysis was performed by integrating BayesSpace data into the Seurat workflow.

**Pseudotime analysis.** Pseudotime analysis was performed using R packages Monocle2 and Monocle3[29,30]. Instead of ordering cells depending on their gene expression profile, we let the algorithm sort pixels based on their spectral information. Once the algorithm has learned the overall "trajectory" of changes, Monocle places each cell/pixel at its proper position in the trajectory. If there are multiple outcomes of this analytical process, Monocle will reconstruct a branched trajectory[51]. As graph learning algorithm we used DDRTree (discriminative dimensionality reduction tree), which is a reversed graph embedding algorithm that allows for dimension reduction while learning the trajectory[30]. Raman spectroscopy data was integrated into Monocle's workflow as demonstrated in Supplementary Fig. 1b.

**Spatial trajectories.** Paths towards the pathologies were described by straight lines. The data matrix was subsetted to pixels passed through on this trajectory. To validate findings of dynamics when approaching pathologies, trajectories were analyzed for approaching pathologies from all four cardinal directions (Supplementary Fig. 11c, d).

**Calculation of dynamics along spatial trajectories.** "Dynamics" of specific Raman wavenumbers describe incline and/or decline of intensities along the spatial trajectory. Quantification of these dynamics was performed by two approaches. We first calculated the maximum difference between the highest value of the local polynomial regression fitting (loess) curve and the lowest value. As this calculating cannot consider variability between these two points, we also calculated the derivation from the loess curve to calculate increase and decrease. All pixels with derivation values $> 0.4$ or $< -0.4$ (meaning $> |0.4|$) were counted and normalized for total number of pixels assigned to the specific cluster along the trajectory. Statistical significance was calculated by Welch Two Sample t-test with R. To analyze reproducibility, we performed experiments in $n = 4$ individual mice and applied paired t-tests when comparing remodeling and healthy clusters within the same mouse.

**Raman-MSI multi-omics to investigate metabolic landscapes.** Raman and MALDI imaging were performed on directly adjacent sections freshly cut from the histo-blocks. The Raman scan was performed at a resolution of 10 μm and the MALDI scan at 50 μm. To spatially align both scans we used a landmark assisted approach, combining H&E stainings performed after the MALDI scan and brightfield images of Raman scans. For co-registration of Raman and MALDI pixels we established a custom affine transformation. Briefly, after spatial alignment of both scans the rotation angle, scaling factor and translation of the Raman scan were defined. Next, the center of each MALDI scan pixel was calculated and the spatially corresponding Raman scan pixel was calculated by the custom affine transformation. Both datasets were integrated into one Seurat object as individual assays, imzML files were imported using the R package Cardinal[52]. The basic Seurat workflow was performed separately with both datasets. Multimodal analysis was then performed using weighted nearest neighbor analysis. The clusters found were compared to the H&E stainings to verify location of healthy and ischemic clusters. Next, cluster averages for Raman and MALDI datasets were calculated for each of $n = 3$ individual samples. Volcano plot analysis was applied comparing healthy and ischemic regions with paired t-tests.

**Raman-MACSima™ multi-omics to identify cellular (immune-) landscape.** After selecting the periinfarct region by H&E Staining, Raman scans were performed as described before, at a dimension of $300 \times 300$ pixel resp. μm. To correctly overlay Raman scan and immunofluorescence image, we aligned strong Hemin peaks at 1372 cm$^{-1}$ from erythrocytes with strong autofluorescence of erythrocytes in the immunofluorescence image (Supplementary Fig. 13) and used concise landmarks visible in both images. Next, pixel with cell-specific immunolabeling were manually selected using ImageJ's built-in multipoint tool. To corresponding spectra of the selected pixels were subjected to dimension reduction and plotted using t-distributed stochastic neighbor embedding (tSNE), provided by Rtsne package.

**Deep learning models.** Preprocessed spectra were used to train deep learning models built in Tensorflow[53]. Models were constructed based on the autoencoder architecture, which comprises an encoder-decoder pair. The encoder was responsible for learning a latent feature representation of the input data, enabling the decoder to reconstruct the original input sequence. To reconstruct Raman spectra, three distinct model architectures were trained: (1) a long short-term memory (LSTM) autoencoder[54], (2) a simple U-net[55], and (3) a U-net incorporating LSTM layers. All models were trained to minimize the mean squared error between the input and the reconstructed Raman spectra. Nine Raman scans were used for trainings and three for validation/testing. Additionally, k-means clustering was applied to extracted latent features to identify an equal number of clusters as the Raman Spectromics approach. Subsequently, cluster images were generated for each tissue section and quantitatively compared to cluster images obtained through Raman Spectromics. The evaluation was performed using the intersection over union (IoU), also known as the Jaccard Index, a widely used metric for assessing image segmentation models[56]. To facilitate the comparison, masks were applied to both the Raman Spectromics cluster images and the cluster images obtained from deep learning. For each image, N clusters were present, resulting in the generation of N masks. By computing the IoU between each pair of masks, an IoU matrix of size N by N was constructed. In this matrix, each column corresponds to a Raman Spectromics cluster, while each row represents a cluster derived from the k-means clustering or deep learning models. To simplify the interpretation of the matrices, the maximum IoU value for each column and row was extracted, ensuring that each unique pair of rows and columns contributed only one IoU value. The average of these maximum IoU values indicates the clustering approach that best reproduces the clustering achieved by Raman Spectromics.

**Illustrations.** Microscope illustrations and icons were modified from BioRender.com with a corresponding license and created using PowerPoint.

### Statistics and reproducibility

Quantification data is displayed as mean ± standard error of mean (SEM), or as indicated in each figure legend. All data conformed to the normal distribution. Statistical significance between the two groups was calculated by Welch two-sided t-test with $p$ values $< 0.05$ considered significant. The micrographs and Raman images shown are either representative or images of all analyzed datasets are shown.

### Reporting summary

Further information on research design is available in the Nature Portfolio Reporting Summary linked to this article.

## Data availability

Raw data from Raman analysis as well as images from histological stainings used in this study have been deposited in the Zenodo repository under accession code https://doi.org/10.5281/zenodo.8265653 [https://doi.org/10.5281/zenodo.8265653]. We highly encourage other scientists to test further clustering algorithms on our datasets provided. Additional data can be obtained from the corresponding author upon request. Source data are provided with this paper.

## Code availability

The programmatic workflow is available in the supplements (Supplementary Fig. 1b). Spatially-unaware and integrated cluster analysis was performed using Seurat with setup instructions and vignette available at https://satijalab.org/seurat/index.html. Spatial clustering was performed using BayesSpace with setup instructions and vignette available at https://edward130603.github.io/BayesSpace. Pseudotime Trajectory analysis was performed using Monocle2 and 3 with setup instructions and vignette available at http://cole-trapnell-lab.github.io/monocle-release/. Specific code can be obtained from the corresponding author upon request.

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

## Acknowledgements

This research was supported by funding from the Interdisciplinary Center for Clinical Research Tuebingen (IZKF) Doctoral Program to M.S., the Deutsche Forschungsgemeinschaft (DFG, German Research Foundation) (INST 2388/64-1) and the State Ministry of Baden-Wuerttemberg for Economic Affairs, Labour and Housing Construction (3-4332.62-NMI/65) both to K.S.-L, and the Deutsche Forschungsgemeinschaft SFB/Transregio 205 S01 to A.W. and N.S. High-performance computing was supported by the BMBF-funded de.NBI Cloud within the German Network for Bioinformatics Infrastructure (de.NBI) (031A532B, 031A533A, 031A533B, 031A534A, 031A535A, 031A537A, 031A537B, 031A537C, 031A537D, 031A538A).

## Author contributions

M.S., J.M., and M.P.G. designed the project. M.S. operated the mice for the hypertrophy model, performed the Raman experiments, data analysis and wrote the paper. J. M., M.S., N.S., and A.W. analyzed and interpreted the data. A-K.R. and J.S. operated the mice for the myocardial infarction model. V.H., S.S., and U.G. performed immunofluorescence experiments. N.S. and A.W. performed MALDI Mass Spectrometry Imaging Experiments. M.K. trained and evaluated deep learning models. M.P.G., K.S-L., C.M.S., P.B.M., and D.H. supervised the experiments. All authors discussed the results and commented on the manuscript.

## Funding

## Competing interests

The authors declare no competing interests.

## Additional information

[1]Department of Cardiology and Angiology, University Hospital Tuebingen, Eberhard Karls University Tuebingen, 72076 Tuebingen, Germany. [2]UCD Conway SPHERE Research Group, Conway Institute, University College Dublin, Dublin, Ireland. [3]School of Biomolecular and Biomedical Science, University College Dublin, Dublin, Ireland. [4]Department of Pediatric Hematology and Oncology, University Children's Hospital Tuebingen, 72076 Tuebingen, Germany. [5]Cluster of Excellence iFIT (EXC 2180) "Image-Guided and Functionally Instructed Tumor Therapies", University of Tuebingen, 72076 Tuebingen, Germany. [6]Research Unit Analytical Pathology, Helmholtz Zentrum Muenchen, German Research Center for Environmental Health (GmbH), Neuherberg, Germany. [7]Institute of Biomedical Engineering, Department for Medical Technologies and Regenerative Medicine, Eberhard Karls University Tuebingen, 72076 Tuebingen, Germany. [8]NMI Natural and Medical Sciences Institute at the University of Tuebingen, 72770 Reutlingen, Germany. [9]Institute for Discovery, O'Brien Centre for Science, University College Dublin, Dublin, Ireland. [10]These authors contributed equally: Julia Marzi, Meinrad Paul Gawaz. ✉e-mail: Meinrad.Gawaz@med.uni-tuebingen.de

