## [Peer Review File · Nature Communications]

Translating genomic tools to Raman spectroscopy analysis enables high-dimensional tissue characterization on molecular resolutionREVIEWER COMMENTS

Reviewer #1 (Remarks to the Author):

The manuscript by Sigle et al. reports about a Raman tissue image analysis workflow. Precisely the authors implemented analysis procedures or specifically software packages developed for transcriptomics to analyze Raman spectra. I do not have much to criticize on the presented results, but I think the authors oversell their results. In my opinion this is yet another study which can be summarized as “Raman spectral histopathology”. Raman spectroscopy or more precisely hyperspectral Raman imaging has been used for a long time to study tissue sections. The most crucial point in this respect is the analysis of the hyperspectral Raman data cubes by means of sophisticated chemometric or machine learning approaches. Numerous studies are available reporting about novel Raman analysis routines towards Raman spectral histopathology. In my opinion the manuscript presented by the authors fits seamlessly into these studies.

Here, the authors used software packages applied in transcriptomics to analyze Raman spectra. First of all, it is not clear to me what methods exactly are implemented within “Seurat” or “Monocle”. Secondly the presented analysis results do not convince me to be superior to what is known in literature in terms of analyzing Raman images by modern machine learning approaches. Thirdly I do not understand why the authors emphasize so much the aspect of “spatially resolved spectral information”, which is also used in other analysis routines. Spectral clustering to retrieve spatiomolecular information is known. Thus, the results section reads like any other results section of other manuscripts analyzing Raman spectroscopy by tailored machine learning routines. I do not see how the presented approach is superior to any other of these commonly applied chemometric or machine learning approaches. What new information can be exactly gained not accessible before? What would be really new and innovative would be the combination of Raman imaging and spatially resolved transcriptomics.

One example on page 7 the authors write “These results confirm that univariate, intensity-based readouts are insufficient for precise identification of biological tissue structures and support the hypothesis, that in-depth analysis and interpretation of complex hyperspectral imaging data requires multivariate data processing. Also, molecular sensitive spatial assignments rather rely on holistic spectral patterns than on single peaks”. This is known for a long time and is not surprising at all.

The fact that hyperspectral Raman images correlate well with pathological staining is also well known.

The authors also used different transcriptomic methods to improve their results. This is also known from spectral histopathology that a preselection of spectral clusters by one model and the analysis of the preselected clusters by another model leads to an improved analysis outcome.

Furthermore, the authors write on page 10: “As noticed before, we were surprised that the clustering algorithms could find myocardial zonations, which classical staining could not uncover”. This is also not surprising since within the hyperspectral Raman images more information is encoded than visible in pathological stainings. From IR-Imaging it is known that hyperspectral IR images can be translated into various different stainings simultaneously.

The discussion is written in very general terms and reads like many other motivations trying to show the advantages of Raman spectroscopy for labelfree analysis of tissue.

To sum up, the authors used omics analysis tools to analyze Raman spectra and called this Raman spectromics. In my opinion this oversells the results since numerous other Raman spectromics studies exist. The term Raman spectromics should in my opinion be used more

generally and should apply to any study using Raman spectroscopy to study cells and tissue. The authors are correct by making the comparison to omics in stating that “wavenumbers can be interpreted like a gene transcript” and all other publications using Raman spectroscopy to analyze wavenumbers via standard chemometrics or more sophisticated machine learning methods can be considered as Raman spectromics too. For me the advantages of utilizing spatial transcriptomic analysis to Raman data analysis as compared to all the other Raman analysis approaches is not obvious at all (I assume the transcriptomic software packages are also based on machine learning!). I do not see any aspect reported in the manuscript which could not be revealed by machine learning approaches to analyze histological samples. Thus, I think this study can be published in a specialized journal but is not of interest for Nature communication dedicated to a rather broad readership. One last comment, the authors used deparaffinized samples which are to my opinion not really well suited for Raman spectroscopy since the lipids are washed out and the Raman spectra do not contain any lipid information anymore.

Reviewer #2 (Remarks to the Author):

In the submitted manuscript, Sigle et al. describe the use of Raman spectroscopy for multidimensional spatial analysis of cardiac tissue. This approach involves the acquisition of spectral data obtained after exposing tissue to a laser within small regions of interest using a confocal microscope. The authors employ mouse models of myocardial infarction and pressure overload (AngII infusion in ApoE^{-/-} mice) to examine the utility of this approach. Using single cell RNAseq packages to analyze their data, they find that Raman spectroscopy identifies spatially restricted features that appear to localize within biologically distinct myocardial regions that contain distinct cell types and extents of fibrosis.

This study examines an interesting technology that may add to the emerging compilation of techniques for spatial tissue profiling. However, there are a number of limitations that limit the applicability of this modality and the validity of the author’s conclusion. Key concerns include the limited number of features extracted from the data, unclear identify of spectral signals, lack of direct comparison to other spatial profiling techniques, and numerous conclusions that are not supported by the presented data.

Major comments:

1. The major limitation of this study is that it remains unclear if Raman spectroscopy adds biologically meaningful information over traditional histology techniques. Many of the features identified were suggested to correspond to readily identifiable structures: fibrosis, myocardium, RBCs, immune cells, blank areas of slides. No direct comparisons are made to other available spatial profiling techniques such as multiplex immunostaining or spatial transcriptomics. It would be important to know whether Raman spectroscopy adds to information obtained by available methodologies.
2. Inter-sample and intra-sample reproducibility needs to be assessed. How consistent is the peak footprint between different slides obtained from a single sample and individual samples within an experimental condition? Information about data variability is essential to evaluate the usefulness of this approach.
3. A drawback of this approach is the limited number of features extracted from the data and

unclear identify of spectral signals. How are the identity of spectral signals determined? Is it possible to pool spectral signals obtained from a signal cell to increase the information extracted. Was cell segmentation performed? It may be more optimal to present the data at the cell vs spot resolution.

4. The authors do not provide sufficient information regarding their analytical pipelines. How did they decide which peaks were outliers? How was the data normalized and scaled (which method)? How did the authors select a clustering resolution? How many highly variable features were used? How was the robustness of sample integration examined?

5. Many conclusions are made that are not supported by the presented data. The biological meaning of the extracted Raman spectroscopy data is not validated, and thus, conclusions regarding spatially restricted biological processes is not justified. For example, predicted areas of fibrosis should be validated through quantitative and statistical comparisons of predictions and ground truth measurements.

Reviewer #3 (Remarks to the Author):

In this manuscript, the authors aim to demonstrate that defining the spatial distribution of biomolecules by spatially resolved Raman microspectroscopy may allow studying changes in tissues and cells at subcellular resolution.

The authors should be praised for having performed elegant work by integrating established bioinformatics tools for single-cell genomics and spatial transcriptomics with Raman spectra. In particular, the pseudo-time analysis seems promising.

They have applied the methodology to study paraffin-embedded sections from experimental models of myocardial infarction and cardiac hypertrophy.

Although technically impressive, the work proposed is still very preliminary. They have clearly demonstrated the potentialities of their “spectromics” approach for spatially resolved, untargeted Raman spectroscopy. However, data are mainly qualitative and several points still need to be fully investigated to validate the proposed approach:

1. If I understand correctly, the area analyzed ranges from 250 by 250 μm to 300 by 300 μm . How is this area representative of the whole diseased organ?
2. Can the approach be applied to analyze more ROIs in one section? And eventually sequential sections?
3. How long the scan of one single ROI takes?
4. And the subsequent analysis?
5. How is the approach translatable to other labs?
6. The authors compare qualitatively healthy vs. diseased tissues. But what about a quantitative comparison of well-powered groups of mice?
7. How do the different immune clusters identified in Figure 6 fit with flow cytometry or single cell and/or single nuclei RNA-seq data published in similar models/tissues? For example, I see only neutrophils; however, myeloid cells are missing.

Response to referees

We are very grateful to the reviewers for providing such valuable, thoughtful, and thorough feedback. The comments and questions raised have prompted us to strengthen our findings in new and interesting ways and we hope that you are satisfied with our revision. Thank you for your time.

Reviewer #1 (Remarks to the Author):

The manuscript by Sigle et al. reports about a Raman tissue image analysis workflow. Precisely the authors implemented analysis procedures or specifically software packages developed for transcriptomics to analyze Raman spectra. I do not have much to criticize on the presented results, but I think the authors oversell their results. In my opinion this is yet another study which can be summarized as “Raman spectral histopathology”. Raman spectroscopy or more precisely hyperspectral Raman imaging has been used for a long time to study tissue sections. The most crucial point in this respect is the analysis of the hyperspectral Raman data cubes by means of sophisticated chemometric or machine learning approaches. Numerous studies are available reporting about novel Raman analysis routines towards Raman spectral histopathology. In my opinion the manuscript presented by the authors fits seamlessly into these studies. Here, the authors used software packages applied in transcriptomics to analyze Raman spectra.

First of all, it is not clear to me what methods exactly are implemented within “Seurat” or “Monocle”.

Response:

Thank you very much for having reviewed our manuscript and the constructive criticism. Concerning the methods implemented within “Seurat” or “Monocle” we apologize for the lack of clarity. “Seurat” and “Monocle” are widely accepted bioinformatic packages that are being used intensively for analysis of single-cell transcriptomics data. For a more in-depth explanation on how their methods work, we would like to refer to the vignette of these R packages, which are available online under <https://satijalab.org/seurat/index.html> and <http://cole-trapnell-lab.github.io/monocle-release/>.

We translated methods from “Seurat”, “Monocle” and other tools to Raman spectroscopy data. The programmatic workflow including the exact methods implemented by “Seurat” and “Monocle” is already shown in former Supplementary Figure S1. We have now supplemented these workflows by more details, including procedures for outlier removal, quality check (QC) and determination of cluster resolution (see updated **Supplementary Figure S1** or **Figure R1** below).

a

Data preprocessing:

Seurat workflow:

BayesSpace workflow:

Monocle2 workflow:

Monocle3 workflow:

Figure R1: Workflow overview. Overview on data preprocessing procedures and methods implemented by Seurat, BayesSpace and Monocle.

Comment:

Secondly the presented analysis results do not convince me to be superior to what is known in literature in terms of analyzing Raman images by modern machine learning approaches.

Response:

We are thankful for the constructive criticism and would like to emphasize the actual novelty of our findings:

Previous works about machine learning on Raman spectroscopy have used basic PLS or PCA analysis to differentiate between spectra and identify molecular differences of samples. Analysis has mostly not gone beyond cluster analysis and feature extraction¹.

Others have implemented artificial neural networks (CNN, DNN, RNN, GAN)². In these cases, a model is trained with training data and then this model is used to classify an unknown test

data set. Hence, well-characterized training data is needed and limits unbiased tissue characterization or identification of previously unknown features.

Our analyses take a different approach. To sum up the three most decisive findings:

- 1) We integrated Raman data into well-established analytical pipelines widely used by bioinformaticians around the world, which allows best reproducibility and high bioinformatic quality control. Furthermore, downstream combination with other spatial profiling techniques is made possible, as also demonstrated by us.
- 2) We integrated spatial information of Raman scans (i.e. exact location of specific spectra) into the analytical workflow (using BayesSpace), as also performed by state-of-the-art analytical pipelines for spatial transcriptomics/proteomics etc.
- 3) We integrated Raman data into more sophisticated applications of machine learning, like pseudotime analysis for detection of matrix heterogeneity, or resolution enhancement (please see response to Reviewer 3). Extensive quality check methods were also introduced, like provided by “Seurat” and “clustree” (see response to Reviewer 2).

Altogether, our approach is far beyond machine learning approaches currently established. Taking advantage of widely available bioinformatic tools also increases acceptance and understanding for the broad readership.

Comment:

Thirdly I do not understand why the authors emphasize so much the aspect of “spatially resolved spectral information”, which is also used in other analysis routines. Spectral clustering to retrieve spatiomolecular information is known. Thus, the results section reads like any other results section of other manuscripts analyzing Raman spectroscopy by tailored machine learning routines. I do not see how the presented approach is superior to any other of these commonly applied chemometric or machine learning approaches. What new information can be exactly gained not accessible before? What would be really new and innovative would be the combination of Raman imaging and spatially resolved transcriptomics.

Response:

We are grateful for the opportunity to better clarify our rationale behind emphasizing the underrated value of spatial resolution. We agree that in general, every Raman scan acquires spatiomolecular information (= spatially resolved), i.e. Raman spectra and their spatial localization. However, to the best of our knowledge, spatial information has been used ineffectively, if at all: Usually, cluster analysis via dimension reduction has been performed of spectra detached from their location, as this additional information can badly be integrated into a classical PCA containing only spectral information.

In this study, we used BayesSpace to integrate spatial information into the classical cluster analysis (= spatially aware). As was the case for spatial transcriptomics, additional spatial information preserves the spatial context of cells and matrix from the biological point of view. From the bioinformatic perspective, spatially neighboring spectra are more likely to be of the same material and hence, outliers or artifacts can be compensated. Additionally, we did not only use spatial information to improve cluster analysis, but we also introduced spatial trajectories on Raman scans, to identify molecular dynamics within a dataset. We admit, that at the stage of this proof-of-concept-study the full potential of spatial analysis remains unclear,

but we would like to address them in future projects. We admit that the wording “spatially resolved” and “spatially aware” may be confusing in this context. We explained the meaning of the terminology in more detail in the manuscript main text.

We appreciate your suggestion to combine Raman imaging with Spatial Transcriptomics. We followed your idea but – although technically possible – used Spatial Metabolomics instead. As both methods are based on analysis of biomolecules rather than genes, information obtained by Spatial Metabolomics can be better linked to what is analyzed by Raman spectroscopy. We also used this approach to validate and reproduce our findings of metabolic alterations in the infarct area in murine sections of myocardial infarction. The following figure (**Figure R2**) is now part of the revised **Figure 5**. Both methods were used to analyze the same heart region (transition zone from healthy to ischemic myocardium), both acquired datasets were analyzed separately and integratively via weighted nearest neighbor analysis.

Figure R2: High-dimensional characterization of metabolic alterations in myocardial infarction by multimodal Raman-MALDI imaging. **g** Illustration of the multi-omics approach of combining Raman and MALDI imaging on the same tissue regions. For best fitting, resolution of Raman imaging was 5 times higher than MALDI imaging (10 vs. 50 μm). **h**, left Cluster analysis of both datasets separately. Both methods uncover distinct spatially organized clusters at the infarct border region. **h**, right Cluster analysis of overlaying data points after spatial co-registration of Raman and MALDI scans. **i** Integration of these datasets into a multimodal analysis using weighted nearest neighbor (wNN) dimension reduction and cluster analysis (top). The resulting cluster image (bottom) fits well with the boundaries of the infarcted region identified in the H&E staining (Supplementary Figure XXX). **j,k** Volcano plot of $n = 3$ individual hearts exploring spectral differences from cluster-identified I/R border regions vs. remote (healthy) regions. Paired t-test. **l** Average Raman spectrum for I/R border regions and remote regions. **m** Heatmap of standardized mean difference between I/R and remote regions from $n = 3$ individual hearts. Raman-only section with modalities only identifiable by Raman spectroscopy, MALDI-only those using MALDI imaging. Raman + MALDI show reproducibility of measurements across the different methods.

Comment:

One example on page 7 the authors write “These results confirm that univariate, intensity-based readouts are insufficient for precise identification of biological tissue structures and support the hypothesis, that in-depth analysis and interpretation of complex hyperspectral imaging data requires multivariate data processing. Also, molecular sensitive spatial assignments rather rely on holistic spectral patterns than on single peaks”. This is known for a long time and is not surprising at all. The fact that hyperspectral Raman images correlate well with pathological staining is also well known.

Response:

We apologize for the redundant information and improved this section in the manuscript text. We can confirm that this is no novelty. However, we would like to underline that this is also not the read-out of the presented work. Pathological stainings were used as ground truth validation of findings which were identified by unsupervised machine learning. We used classical and immunohistochemical/immunofluorescence stainings, as these are supposed to be widely known and accepted by the broad readership of *Nature communications*.

Comment:

The authors also used different transcriptomic methods to improve their results. This is also known from spectral histopathology that a preselection of spectral clusters by one model and the analysis of the preselected clusters by another model leads to an improved analysis outcome.

Response:

We apologize for the lack of clarity. For the analyses we did not use a biased preselection of any clusters. Cluster sizes were determined by careful inspection of resulting spatially reordered cluster image and identification of the influence of different clustering resolutions on the cluster size using the “clustree” package (please see revised **Supplementary Figure S3**). Moreover, we did not analyze preselected clusters by another model. We analyzed the same dataset by Seurat and BayesSpace which implement two different clustering algorithms (Seurat for spatially-unaware und BayesSpace for spatially-aware clustering), and both pipelines found similar clusters. Only for pseudotime analyses, we selected the two myocardial subclusters found by Seurat, but this was done to visualize transitions from healthy to diseased myocardium specifically restricted to these two clusters.

Comment:

Furthermore, the authors write on page 10: “As noticed before, we were surprised that the clustering algorithms could find myocardial zonations, which classical staining could not uncover”. This is also not surprising since within the hyperspectral Raman images more information is encoded than visible in pathological stainings. From IR-Imaging it is known that hyperspectral IR images can be translated into various different stainings simultaneously.

Response:

We need to apologize for the confusion occurring from methodological novelties and those concerning the cardiac injury mouse models.

We were not surprised by the fact that Raman imaging itself identified the different myocardial zonations, but by the fact that an unbiased clustering algorithm found these subclusters. These clusters (pre-fibrotic boundary around fibrosis, remodeling myocardium) have not yet been described before, but could be reproduced with $n = 4$ independent mice (please see Response to Reviewer 2 and new **Supplementary Figure S8**). We are convinced that the clusters would have not been found by classical PCA or other machine learning approaches, as e.g. information of spatial organization would miss.

Comment:

The discussion is written in very general terms and reads like many other motivations trying to show the advantages of Raman spectroscopy for label-free analysis of tissue.

Response:

We have to admit that the discussion remained partially superficial and does not fully cover in-depth discussion and validation of the implemented technology itself. We tried to specify our findings and to improve the discussion. Please read the revised version of the discussion in the manuscript main text.

Comment:

To sum up, the authors used omics analysis tools to analyze Raman spectra and called this Raman spectromics. In my opinion this oversells the results since numerous other Raman spectromics studies exist. The term Raman spectromics should in my opinion be used more generally and should apply to any study using Raman spectroscopy to study cells and tissue. The authors are correct by making the comparison to omics in stating that “wavenumbers can be interpreted like a gene transcript” and all other publications using Raman spectroscopy to analyze wavenumbers via standard chemometrics or more sophisticated machine learning methods can be considered as Raman spectromics too. For me the advantages of utilizing spatial transcriptomic analysis to Raman data analysis as compared to all the other Raman analysis approaches is not obvious at all (I assume the transcriptomic software packages are also based on machine learning!). I do not see any aspect reported in the manuscript which could not be revealed by machine learning approaches to analyze histological samples. Thus, I think this study can be published in a specialized journal but is not of interest for Nature communication dedicated to a rather broad readership.

Response:

We are sorry that the impression has arisen that we oversell our results and that you find it difficult to see the added value of our study. To support the impact of our work we'd like to clarify some points:

- 1) In our opinion, “omics” is not only applying a dimension reduction technique on a Raman scan. We introduced the term “omics” to Raman spectroscopy to emphasize a comprehensive, holistic, systems-level analysis of all available data acquired by a Raman scan or integrative analysis of Raman data and another omics discipline. The omics techniques demonstrated in this study go beyond current approaches. We do not raise the claim that the methods here are superior to other machine learning approaches. However, translating existing well-characterized and validated tools into

another context add the great benefit of resource-efficiency, transparency, reproducibility, and interpretability.

- 2) We also used the term “omics” to use a “familiar” concept appealing to a broad readership. Another aim of our study is to demonstrate that existing tools can be creatively translated into different contexts without the need to “reinvent the wheel”. This study should not only address scientist working with Raman spectroscopy, we’d like to arouse interest in every reader to translate omics techniques to their field of research. As an example, the R package Seurat has also been used by Crainiciuc et al. to study behavioural immune landscapes of inflammation using 4D immunofluorescence images (*Nature*, 2022)³.
- 3) Machine learning models are currently bursting into the scientific world like a flood. Especially those using deep learning techniques can be considered as “black boxes”, meaning that it is difficult to understand how the model arrived at its prediction. Also training of these models with different samples among labs make reproducibility hard. Other machine learning approaches lack interpretability and transparency. Without having a gold standard, it is impossible to evaluate which machine learning model is the best. In this context, our described method could represent a turning point by offering high transparency, reproducibility and interpretability and hence fulfilling quality criteria for a gold standard.

Comment:

One last comment, the authors used deparaffinized samples which are to my opinion not really well suited for Raman spectroscopy since the lipids are washed out and the Raman spectra do not contain any lipid information anymore.

Response:

We thank the reviewer for this valuable and justified comment. In the present study, we didn’t focus on lipid alterations in the two animal models, so FFPE tissue was the simplest way to handle our hypotheses. For analysis of extracellular matrix, fibrosis and cardiomyocytes this was most adequate. From our experience and also experiments performed for this revision, we noticed a technical benefit of FFPE over cryosections: When performing large area scans, the tissue of cryosections tends to swell, which can cause the confocal laser to get out of focus.

However, cryosections can also be used for our approach without problems. For the revised **Figure 6** we used cryosections, as these also bring the advantage of better immunofluorescence signals. The figure shows the combination of Raman spectroscopy with consecutive MACSima® multiplexed immunofluorescence staining of the identical section. Please find the revised **Figure 6** in the manuscript or as response to Reviewer 2.

Reviewer #2 (Remarks to the Author):

In the submitted manuscript, Sigle et al. describe the use of Raman spectroscopy for multidimensional spatial analysis of cardiac tissue. This approach involves the acquisition of spectral data obtained after exposing tissue to a laser within small regions of interest using a confocal microscope. The authors employ mouse models of myocardial infarction and pressure overload (AngII infusion in ApoE^{-/-} mice) to examine the utility of this approach. Using single cell RNAseq packages to analyze their data, they find that Raman spectroscopy identifies spatially restricted features that appear to localize within biologically distinct myocardial regions that contain distinct cell types and extents of fibrosis.

This study examines an interesting technology that may add to the emerging compilation of techniques for spatial tissue profiling. However, there are a number of limitations that limit the applicability of this modality and the validity of the author's conclusion. Key concerns include the limited number of features extracted from the data, unclear identify of spectral signals, lack of direct comparison to other spatial profiling techniques, and numerous conclusions that are not supported by the presented data.

Major comments:

1. The major limitation of this study is that it remains unclear if Raman spectroscopy adds biologically meaningful information over traditional histology techniques. Many of the features identified were suggested to correspond to readily identifiable structures: fibrosis, myocardium, RBCs, immune cells, blank areas of slides. No direct comparisons are made to other available spatial profiling techniques such as multiplex immunostaining or spatial transcriptomics. It would be important to know whether Raman spectroscopy adds to information obtained by available methodologies.

Response:

We thank the reviewer for the justified objection. One of the major motivations to implement Raman imaging is to support and enhance decision making in pathology. Compared to conventional histological staining, the marker-independent data generation by Raman imaging does not require extensive sample processing and is accessible in various sample stages such as FFPE section, cryosection or even fresh tissue. Moreover, combined with an endoscopic setup, first studies have proven the feasibility of *in vivo* application which could decrease the necessity of tissue biopsies and allow for intraoperative diagnostics⁴. Furthermore, spectral features do not only allow to discriminate tissue structures but can also access submolecular/structural alterations that are not accessible by staining methods; e.g. one of our recent works showed the sensitivity to discriminate native from fibrotic collagen I fibers in human tissue samples⁵. We have supplemented the discussion of the main manuscript text with these important points.

We introduced Raman "Spectromics" as a method for spatially resolved characterization of tissue at molecular resolution. Among other spatial profiling techniques, the most important ones are Spatial Transcriptomics, Spatial Proteomics and Spatial Metabolomics⁶⁻⁸. While Raman spectroscopy can also be applied *in vivo*, the other methods can mostly or exclusively be used *ex vivo*. Raman spectroscopy is also method to require the least tissue preparation. Furthermore, it provides one of the highest resolutions among these techniques. We were asked for direct comparisons to other spatial profiling techniques and would like to demonstrate the results.

Although we already applied multiplexed immunostaining using the MACSima™ platform (former **Figure 6**), we have to admit that the small number of antibodies used in the current approach is not sufficient to be already called multiplexed. The limited number of antibodies resulted from the use of FFPE instead of cryosections, which often show better results. However, to demonstrate the feasibility of combining multiplexed immunostaining with Raman spectroscopy, and also to compare both spatial profiling techniques, we operated more mice to induce myocardial infarction and generate cryosections. A 23-color-panel (**Supplementary Table S2**) was then applied on the tissue. By overlaying immunofluorescence stainings and the Raman scan, immunofluorescence-positive pixels were selected. The revised **Figure 6** shows the results of our new analysis. The previous Figure 6 was moved to the supplementary information and extended by cell vs. spot resolution analysis, as outlined in a response below.

Figure 6

Figure 6 Defining the surrounding cellular (immune-) landscape in the model of acute myocardial infarction. **a** Multi-omics approach to decipher spectra from cells by combination of Raman spectroscopy with consecutive MACSima™ multicolor immunofluorescence staining. **b** H&E staining of murine myocardial infarction subjected to analysis. Scalebar 300 μm . **c** MACSima™ multiplexed immunofluorescence imaging was performed on the same region as Raman spectroscopy previously on the same section. Scalebar 50 μm . **d** Raman Intensity images at 2940 cm^{-1} (general band for lipids and proteins). Scalebar 100 μm . **e** Cluster image identified by the Seurat workflow. Cluster 0 was identified as myocardium. **f** tSNE plot after dimensionality reduction of Raman spectra of identified cell types. Clear separation into cardiomyocytes, erythrocytes and immune cells. **g** Spectral subphenotyping of immune cells. Neutrophils display the largest cluster while MHC II+ professional antigen-presenting cells (pAPCs) and CD68+ macrophages separate into a distinct cluster. **h** Donut chart showing frequency distribution of identified cells (percent from absolute number of analyzed number of pixels). **i** Venn diagram of surface markers analyzed for the immune cell subpopulation. E.g., most Ly6G+ cells were also CD45+. **j** Average spectra for the different cell types and characteristic peaks, together with the spatial representation of the analyzed pixels. vSMC: vascular smooth muscle cells

Besides the multiplex immunostaining we also made an approach combining and comparing Raman Spectroscopy with Spatial Metabolomics, which from our point of view makes more sense than Spatial Transcriptomics, as suggested by Reviewer 1 (please see the response to the reviewer there). Integrating both Raman and MALDI-FT-ICR mass spectra of the same location into one multidimensional analysis could not only strengthen the overall results, but also help both methods to profit from benefits of the other. Please see the **revised Figure 5** for the results.

Comment:

2. Inter-sample and intra-sample reproducibility needs to be assessed. How consistent is the peak footprint between different slides obtained from a single sample and individual samples within an experimental condition? Information about data variability is essential to evaluate the usefulness of this approach.

Response:

We appreciate this valuable suggestion.

We performed experiments to evaluate intra-sample reproducibility by trying to replicate clustering results of adjacent histological sections obtained from a single sample. Sequential scans were performed at the same spatial location of three following sections. Subsequently, multidimensional analysis was performed using similar sizes for dimensions and resolution. Where necessary, cluster order was matched to those found in the first section, as Seurat automatically orders clusters by size. Cluster proportions (i.e. number of pixels resp. spectra) were plotted and display both visually and quantitatively comparable results (please see **(Supplementary Figure S4a-d** and below). We performed the same experiment with $n = 3$ sections across experimental conditions (NaCl control group, **Supplementary Figure S4e** and below and Ang II in **Supplementary Figure S4f**), which also displayed similar results of reproducibility. The findings are promising, and we included a paragraph about reproducibility within the manuscript main text with reference to the new supplementary figure.

Supplementary Figure S4 Evaluation of intra-sample reproducibility. **a** To evaluate the robustness of Raman Spectromics analyses, sequential scans of adjacent sections were performed at the same spatial location. **b** Seurat-derived cluster analysis displays a visually comparable result. The pink cluster from Section A and the dark green cluster from B and C derive from paraffin residues on the slides. **c** Quantitative analysis of cluster sizes by direct comparison of the cluster proportions, underlining reproducibility in different sections. **d** Cluster average spectra found in all three samples. While Cluster 0 to 4 display a high fit, cluster 5 appears more heterogenous. Looking at the spatial distribution of the red cluster, one can consider it as artifact cluster. **e/f** Further sections of the control group (**e**) and another area of subendocardial fibrosis in the Ang II treatment group (**f**). Cluster proportions show comparable results. In section B from **Figure 2f** an artificial cluster occurred from a tissue fold in the scan area.

We also performed experiments to evaluate inter-sample reproducibility.

For the hypertrophy model, we replicated the finding of a distinct “remodeling myocardium”-subcluster in further sections of $n = 4$ individual mice, as displayed in the new **Supplementary Figure S8**. Remodeling myocardium was found to be in close neighborhood to fibrotic areas, while the light blue healthy myocardium is far away. We also performed PCA on the spectra of the corresponding clusters. Visual inspection and quantitative analysis confirmed significant differences between healthy and remodeling myocardium.

Supplementary Figure S8: Inter-sample reproducibility of cluster found for remodeling myocardium. **a** Cluster images of 4 individual hearts with areas of fibrosis and surround area of remodeling myocardium. Pink denotes the fibrotic cluster; light blue is interpreted as healthy and dark blue remodeling myocardium. **b** Principal Component Analysis (PCA) of the clusters found by unsupervised cluster analysis in **a**. Spectra for healthy and remodeling myocardium show a strong separation into 2 clusters. **c** Loadings plot for Principal Component 1 (PC1). All 4 samples show comparable spectral and hence molecular differences leading to two distinct myocardial subclusters. **d** PC Scores plot for PC1 and PC. PC1 displays highly significant differences between healthy and remodeling myocardium ($p = 0.0002$), PC4 is nearly significant ($p = 0.0832$). $n = 4$ from 4 individual mice. Paired t-test.

Please also see our responses to Reviewer 3, who also asked for statistical analysis of well-powered groups. We provided these measures of reproducibility for both mouse models and all findings. The results can now be found in the revised **Figure 4** as well as **Supplementary Figure S8 and S9**.

When combining Raman spectroscopy with Spatial Metabolomics, we also used high-resolution mass spectrometry to validate our metabolic findings. Please find the new results in the revised **Figure 5**.

Comment:

3. A drawback of this approach is the limited number of features extracted from the data and unclear identity of spectral signals. How are the identity of spectral signals determined? Is it possible to pool spectral signals obtained from a signal cell to increase the information extracted. Was cell segmentation performed? It may be more optimal to present the data at the cell vs spot resolution.

Response:

We thank the reviewer for his justified objection. It is true that at the current stage, interpretation of Raman spectra is based on our inhouse or literature reference spectra, or feature validation by conventional stainings or immunofluorescence stainings. As for Figure 6 you are referring to, we used immunofluorescence stainings to detect inflammatory cells at the side of acute myocardial infarction. To this end we created a pixel mask derived from the Raman scan and overlaid the immunofluorescence staining with this mask. Next, all immunofluorescence-positive pixels were selected using an inhouse ImageJ script. Subsequently corresponding spectra to the positive pixels were extracted from the Raman scan and multidimensional analysis was performed together with spectra from other cell types.

As you suggested, we also compared this “spot resolution” or in our case “pixel resolution” by cell resolution. For this purpose, we performed cell segmentation and pooled all pixels from one cell. Next, we performed multidimensional analysis with these pooled data. The results using this “cell resolution” with pooled Raman data is clearly inferior to the “pixel resolution” approach (please see the new **Supplementary Figure S14** or **Figure R3** below), as dividing into distinct clusters was reduced. The problem of pooling and averaging multiple spectra by doing cell resolution is that eventual outliers spectra are merged to actual correct spectra. The resulting average spectra is composed by true and false spectra, which make multidimensional analysis incorrect.

Figure R3 Cell vs. pixel resolution analysis. **a** Immunofluorescence images were used to segment neutrophils and vascular cells. **b** Selected neutrophils and myofibroblasts/VSMC for analysis. Average spectra for each cell were calculated at first and then subjected to cluster analysis. **c** Linear (PCA) and non-linear (tSNE) dimension reduction show a weak separation into clusters for both cell types. **d** Average spectra from cell (colored) vs. pixel (in grey) resolution. Main differences between both spectra are labeled with the corresponding wavenumber. Differences are the result of outlier spectra which have been merged into actual correct spectra.

Comment:

4. The authors do not provide sufficient information regarding their analytical pipelines. How did they decide which peaks were outliers? How was the data normalized and scaled (which method)? How did the authors select a clustering resolution? How many highly variable features were used? How was the robustness of sample integration examined?

Response:

We apologize for the lack of clarity and would like to explain more in detail. We have completed the methods section and the programmatical workflow (**Supplementary Figure S1**) with more detail (see response to Reviewer 1).

Outliers were defined as spectra with an intensity difference > 0.005 A.U between two adjacent wavenumbers, as visualized exemplary in **Figure R4**. If this was the case, the whole spectrum was excluded from analysis (and not a single peak within this spectrum). Summarizing all Raman scans for this publication, we removed $\emptyset 0,024\%$ outlier spectra per scan (range 2-86 pixels per 62500 pixels). Outliers can be found as white spots in der spatially reordered cluster images.

Figure R4 Outlier detection. Intensities > 0.005 A.U. within a spectra caused the whole spectrum to be filtered out. For visualization purpose only 1,000 spectra were randomly selected from the Raman scan, containing 62,5000 spectra.

Data was normalized by the normalization function “Area to 1” provided by Project FIVE, which is proprietary software from the vendor of the Raman microscope and normalizes the spectrum to overall spectral area under the curve to 1 [a.u.].

No further normalization was performed in Seurat. After normalization, data was scaled, meaning that each feature (i.e. Raman wavenumber) is centered to have a mean of 0 and is scaled by the standard deviation of each feature.

Clustering resolution was evaluated using the tool clustree (<https://cran.r-project.org/web/packages/clustree/>), which is also recommended by the developers of Seurat. The final cluster resolution was selected by careful consideration of what makes biological sense, to avoid over- or underclustering. For the analysis of the scans in this manuscript, clustering resolution was between 0.2 and 0.5. The revised **Supplementary Figure S3** visualizes the clustree analysis exemplary for the section of subendocardial fibrosis.

Supplementary Figure S3 Influence of cluster resolutions analyzed by “clustree” package. **a** Clustering tree of different resolutions. Selecting a resolution of 0.1 results in 4 clusters (second row, dark yellow). Increasing the resolution to 0.2, results in a split of cluster 1 into 2 subclusters. The 2 clusters to the left are that distinct, that changes in cluster resolution do not influence the value of k clusters. **b** Overlaid clustree analysis with spectra in UMAP projection, demonstrating the split of clusters as visualized by the tree, but here in UMAP projection. **c** Cluster images when using different resolutions.

As highly variable features we used all available wavenumbers in this study. The rationale behind this was to avoid any bias by subsetting data to any specific wavenumbers. However, the selection of n highly variable features could be introduced in the future to improve performance and reduce clustering noise for example.

Robustness of our approach with Raman Spectromics was examined by quantitative and qualitative means. While before submission you would find mainly qualitatively testing of our approach using different ROIs within a sample, using different animal models with different pathologies we have now revised the manuscript by adding valuable information about reproducibility, as also suggest by you before and Reviewer 3.

Comment:

5. Many conclusions are made that are not supported by the presented data. The biological meaning of the extracted Raman spectroscopy data is not validated, and thus, conclusions regarding spatially restricted biological processes is not justified. For example, predicted areas of fibrosis should be validated through quantitative and statistical comparisons of predictions and ground truth measurements.

Response:

We appreciate this valuable and justified comment. For this proof-of-principle study, our main focus was not to extensively validate our novel findings in the murine pathologies on top of validating the feasibility of the approach using Raman spectromics. We hope that you will understand our situation. However, we gave our best to validate findings by other methods. As an example, we used multiplexed immunofluorescence imaging, as described before. For metabolic alterations we confirmed our findings with Spatial Metabolomics.

As you suggested concretely, we also performed ground truth comparisons regarding fibrosis. For this purpose, sections were stained for fibrosis using Picosirius Red staining after the Raman scan. Images were overlaid for ground truth (= Picosirius Red Staining) classification and the accuracy of predicted fibrotic regions was determined by counting the number of correctly and falsely predicted pixels. Accuracy was then calculated by

$$Accuracy = \frac{TP + TN}{TP + TN + FP + FN}$$

where TP = True Positive, TN = True Negative, FP = False Positive, FN = False Negative pixels. The ground truth fibrosis positive pixels were defined as > 50% Picosirius Red area per pixel. Accuracy of prediction of fibrotic regions was between 0.84 and 0.98 (please see the new **Supplementary Figure S6** below). BayesSpace proved to be more robust with less outliers. Confusion matrix analysis revealed that BayesSpace has a higher sensitivity (True Positive Rate, TPR) and specificity (True Negative Rate, TNR), while Seurat was more susceptible to outliers and clustering noise. These results support our previous findings that integrating the spatial resolution into analysis improves clustering results and reduces artifacts.

Supplementary Figure S6 Accuracy of fibrosis predictions using unsupervised clustering algorithms. **a** Picosirius red staining used as ground truth for fibrotic areas. **b** Prediction of fibrotic areas by Seurat. Dark colors represent true positive (TN, dark red) or true negative (TN, dark blue) predictions, while light colors reflect false negative (FN, light red) or false positive (FP, light blue) assignments. **c** Prediction of fibrotic areas by BayesSpace. Overall, both classifications look quite similar and accurate. **d** Quantification of the accuracy of fibrosis prediction. BayesSpace outperforms Seurat and is more robust than Seurat's classification. **e/f** Confusion matrix of Seurat's (left) and BayesSpace's (right) predictions, by calculating the mean rates (± standard deviation) of prediction. The true positive rate (dark red) is also known as sensitivity, and the true false rate (dark blue) is the specificity. BayesSpace provides higher sensitivity and specificity in prediction of fibrotic clusters, or overall cluster assignments.

Reviewer #3 (Remarks to the Author):

In this manuscript, the authors aim to demonstrate that defining the spatial distribution of biomolecules by spatially resolved Raman microspectroscopy may allow studying changes in tissues and cells at subcellular resolution. The authors should be praised for having performed elegant work by integrating established bioinformatics tools for single-cell genomics and spatial transcriptomics with Raman spectra. In particular, the pseudo-time analysis seems promising.

They have applied the methodology to study paraffin-embedded sections from experimental models of myocardial infarction and cardiac hypertrophy. Although technically impressive, the work proposed is still very preliminary. They have clearly demonstrated the potentialities of their “spectromics” approach for spatially resolved, untargeted Raman spectroscopy. However, data are mainly qualitative and several points still need to be fully investigated to validate the proposed approach:

Response:

We were delighted to read that our manuscript has attracted your interest and would like to thank the reviewer for his/her encouraging comments that helped us to improve our manuscript.

Comment:

1. If I understand correctly, the area analyzed ranges from 250 by 250 μm to 300 by 300 μm . How is this area representative of the whole diseased organ?
2. Can the approach be applied to analyze more ROIs in one section? And eventually sequential sections?
3. How long the scan of one single ROI takes?
4. And the subsequent analysis?

Responses:

Ad 1.) We appreciate this valuable and justified comment. In this study, the ROIs are indeed small and may not necessarily be representative for the entire pathology. A scan of a region of 250x250 μm with a resolution of 1 μm needs about 1.5 hours. Understandably, increasing the scan region also increases scan time. However, scan time can be reduced by lowering the resolution or decreasing the integration time (i.e., the time how long the laser rests at a pixel position to acquire a spectrum). Reducing the integration time leads to less sensitive scans and is not recommended, increasing the pixel size might depend on the required tissue resolution. Another approach for reducing scan time and increasing the scan area is CARS (Coherent anti-Stokes Raman Spectroscopy) or SRS (Stimulated Raman Scattering) that enable to actively target defined spectral features (single wavelengths), resulting in a stronger signal intensity and up to video-rate acquisition speed^{9,10}. However, these approaches are rather supervised as the target wavelength needs to be defined ahead and, in comparison to spontaneous Raman imaging, is only restrictedly accessible to multiplexing (e.g. by broadband setups and nanotags^{11,12}).

Besides the technical side, there also exist bioinformatical methods to increase the resolution, for example as proposed by BayesSpace by enhancing subspot resolution with a Bayesian statistical method. However, this approach consumes exceptional high computing power, but as a proof-of-principle, we showed the feasibility of this approach with Raman spectroscopy

data in the following **Figure R5**. Please note that spatial enhancement needs extensive amounts of computation power and was performed on a high performance computing system (de.NBI cloud).

Figure R5 Computational resolution enhancement using BayesSpace. a BayesSpace can increase clustering resolution by calculation of subspot cluster IDs. As proof of feasibility, we scanned sections at high resolution (all squares). Spatially-aware cluster analysis using BayesSpace identified 5 distinct clusters (b, left). Next, all spectra were removed from the dataset except for the grey squares. By this, the resolution was reduced from 1 to 9 μm^2 . Note, spatially-aware clustering analysis was performed again and demonstrated highly comparable results to previously, but at a lower resolution (b, center). The number of clusters was 4 instead of 5 previously, resulting from the lower resolution incapable of detecting erythrocyte/hemin spectra. Finally, the reduced dataset was used again for resolution enhancement (b, right). The fibrotic area now resembles more to that detected by cluster analysis of the high resolution.

Ad 2.) Another approach for tackling the problem of representability is by extending analysis to more ROIs in one section. This is a common practice, and an exemplary result is presented in **Figure R6**. It is also no problem to perform analysis of sequential sections, as presented in response to a question from reviewer 2 in the context of reproducibility (see **Supplementary Figure S4**). For the combination of Raman spectroscopy with Spatial Metabolomics, we performed scans of large regions, but at reduced resolution. You can find the results in the revised **Figure 5**.

Figure R6 Scans of multiple ROIs and whole-heart scans. 4 ROIs of 250x250 μm were consequently scanned in the same section. This approach can be useful to increase representability or compare different compartments within one section, without the need to scan the whole section and still having a high resolution for each ROI. Scalebar on ROIs is 50 μm , scalebar on Picrosiriusred staining is 500 μm .

Ad 3.) As mentioned before, for our standard image acquisition the scan time is about 1.5 hours.

Ad 4.) We appreciate this pragmatic question. We have provided an overview over the steps included in our analyses in the revised **Supplementary Figure S1** (see also responses to Reviewer 1 and 2). Raw data preprocessing needs about 5 min, import into the R environment and performing data wrangling and outlier removal another 5 min. Analysis with Seurat takes less than 5 min. Analysis with BayesSpace, in contrast, lasts significantly longer, with about 30 min computation time to estimate the correct cluster size and another 30 min for cluster analysis. All analyses (excluding high performance computing) were performed on a computer running Windows 10 with 12 GB of RAM and an Intel Core i5-6400 CPU. Using a dedicated workstation PC for Machine Learning with better CPU and RAM could reduce computation time.

Comment:

5. How is the approach translatable to other labs?

Response:

We thank the reviewer for this important question and are pleased for the opportunity to better clarify how other labs can benefit from the presented approach.

Concerning labs that are not familiar with Raman but are interested in multidimensional analyses, our approach could make an impulse on how to translate well-established bioinformatic tools to other contexts. As an example, the package Seurat has already been used in a completed different approach than to genomics, here to study behavioural immune landscapes of inflammation using 4D immunofluorescence images (Crainiciuc et al., *Nature* 2022)³.

Concerning labs that also work with Raman spectroscopy, we implemented detailed information about our workflow (**Figure S1**) on how to integrate Raman data into the Seurat, BayesSpace and Monocle environment. Since all packages are open-source, under continuous development and have a broad intellectual support we are convinced about the usefulness of our approach.

However, one little restriction must be pointed out. As described by Guo et. al¹³, Raman spectroscopy may show variability and differences in sensitivity among different labs. Therefore, the applied biomarkers and reference spectra have to be carefully interpreted and evaluated as they can also be tissue- or species-specific. However, the presented data processing pipeline was exemplarily elaborated for myocardium data, but can easily be adapted to any other kind of hyperspectral Raman images as well as different sample origins.

Comment:

6. The authors compare qualitatively healthy vs. diseased tissues. But what about a quantitative comparison of well-powered groups of mice?

Response:

We thank the reviewer for this valuable comment. The present study demonstrates both methodological novelties and those concerning the cardiac injury mouse models. Our main focus was on establishment and validation of our new approach to Raman Spectromics.

However, you are right that the discovered findings for the cardiac injuries mouse models lack quantitative comparison.

In the context of multi-layered datasets, quantitative comparison of selected variables would not do justice to an unbiased analysis, hence we tried to preserve the multidimensional quality of our dataset.

For the hypertrophy model, we confirmed our findings of a remodeling cluster in further sections of $n = 4$ mice, as also asked by Reviewer 2 (see **Supplementary Figure S8**). Some of these results were integrated into the revised **Figure 4** (see also **Figure R7** below)

Figure R7 **a** A similar myocardial pattern with a distinct healthy (light blue) and remodeling (dark blue) cluster when approaching fibrotic regions (pink) was reproduced in 4 individual hearts. **b** Reproduction of intensity shifts of collagen and cytochrome c ($n = 4$). **c** Violin plots showing significantly lower spectral dynamics of cluster 0 (light blue, healthy myocardium) in comparison to cluster 1 (dark blue), implying molecular dynamics of myocardium under remodeling. Quantification by calculating the maximum variability of the local polynomial regression fitting (loess) curve (left) and thresholding of the derivation from loess curve (see Methods). Welch Two Sample t-test. **d** Reproduction of spectral dynamics in the remodeling subcluster in $n = 4$ individual hearts. Paired t-test.

For the myocardial infarction model, we compared $n = 4$ ligated vs. sham-operated mice. We found differences in cluster numbers, determined by unsupervised cluster analysis of the molecular composition of the specimens. The size of pixels associated to erythrocyte clusters was significantly increased in hearts where the LAD was ligated. Similar results were found for the number of nuclei identified by clusters assigned to DNA/RNA. The results underlining the reproducibility can be found as new **Supplementary Figure S9**.

Supplementary Figure S9 **Inter-sample reproducibility and statistical comparisons of myocardial infarction sections.** **a** Representative H&E Stainings of heart regions in close neighborhood of the ligature (top) or sham-operated region. **b** Cluster images determined by unsupervised cluster analysis based on molecular composition of the sample. **c** Quantification of the cluster number found by Seurat's clustering algorithm. **d** Quantification of the cluster size of clusters resulting from erythrocytes and their characteristic hemine peaks. **e** Quantification of the cluster size of clusters resulting from characteristic peaks for RNA/DNA.

For the analysis of metabolic alteration in the myocardial infarction model, findings were replicated in n = 3 mice and validated by high mass-resolution MALDI FT-ICR-MSI (see revised **Figure 5**).

Comment:

7. How do the different immune clusters identified in Figure 6 fit with flow cytometry or single cell and/or single nuclei RNA-seq data published in similar models/tissues? For example, I see only neutrophils; however, myeloid cells are missing.

Response:

We apologize for the lack of clarity. In the former **Figure 6**, we only stained for neutrophils and not myeloid cells, as immunolabeling of cells in paraffin sections is not easy. Hence, myeloid cells are only technically missing. For this revision, we operated more mice and generated cryosections for multiplexed immunofluorescence staining. For the results, please see the revised **Figure 6** and the response to the question of Reviewer 2.

From literature it is known that the major peak of the immune response in permanent myocardial infarction is around day 5 to 7, with immune clusters predominantly defined by lymphatic cells and macrophages¹⁴. We performed transient ligation of the LAD and sacrificed mice already after 24 hours of ischemia-reperfusion injury, resulting in less tissue damage and a lower immune response. After this time period, neutrophils make up the largest fraction of immune cells¹⁴, as also shown by us (revised **Figure 6**).

References

- 1 Guo, S., Popp, J. & Bocklitz, T. Chemometric analysis in Raman spectroscopy from experimental design to machine learning-based modeling. *Nat Protoc* **16**, 5426-5459, doi:10.1038/s41596-021-00620-3 (2021).
- 2 Luo, R., Popp, J. & Bocklitz, T. Deep Learning for Raman Spectroscopy: A Review. *Analytica* **3**, 287-301, doi:10.3390/analytica3030020 (2022).
- 3 Crainiciuc, G. *et al.* Behavioural immune landscapes of inflammation. *Nature*, doi:10.1038/s41586-021-04263-y (2022).
- 4 Heng, H. P. S., Shu, C., Zheng, W., Lin, K. & Huang, Z. Advances in real-time fiber-optic Raman spectroscopy for early cancer diagnosis: Pushing the frontier into clinical endoscopic applications. *Translational Biophotonics* **3**, doi:10.1002/tbio.202000018 (2020).
- 5 Becker, L. *et al.* Raman Microspectroscopy Identifies Fibrotic Tissues in Collagen-Related Disorders Via Deconvoluted Collagen Type I Spectra. *SSRN Electronic Journal*, doi:10.2139/ssrn.4294703 (2022).
- 6 Moffitt, J. R., Lundberg, E. & Heyn, H. The emerging landscape of spatial profiling technologies. *Nat Rev Genet* **23**, 741-759, doi:10.1038/s41576-022-00515-3 (2022).
- 7 Petibois, C. 3D Quantitative Chemical Imaging of Tissues by Spectromics. *Trends Biotechnol* **35**, 1194-1207, doi:10.1016/j.tibtech.2017.08.002 (2017).
- 8 Aichler, M. & Walch, A. MALDI Imaging mass spectrometry: current frontiers and perspectives in pathology research and practice. *Lab Invest* **95**, 422-431, doi:10.1038/labinvest.2014.156 (2015).
- 9 Ozeki, Y. *et al.* High-speed molecular spectral imaging of tissue with stimulated Raman scattering. *Nature Photonics* **6**, 845-851, doi:10.1038/nphoton.2012.263 (2012).
- 10 Evans, C. L. *et al.* Chemical imaging of tissue in vivo with video-rate coherent anti-Stokes Raman scattering microscopy. *Proceedings of the National Academy of Sciences* **102**, 16807-16812, doi:10.1073/pnas.0508282102 (2005).
- 11 Kim, S. H. *et al.* Multiplex coherent anti-stokes Raman spectroscopy images intact atheromatous lesions and concomitantly identifies distinct chemical profiles of atherosclerotic lipids. *Circ Res* **106**, 1332-1341, doi:10.1161/CIRCRESAHA.109.208678 (2010).
- 12 Nicolson, F. *et al.* Non-invasive In Vivo Imaging of Cancer Using Surface-Enhanced Spatially Offset Raman Spectroscopy (SESORS). *Theranostics* **9**, 5899-5913, doi:10.7150/thno.36321 (2019).
- 13 Guo, S. *et al.* Comparability of Raman Spectroscopic Configurations: A Large Scale Cross-Laboratory Study. *Anal Chem* **92**, 15745-15756, doi:10.1021/acs.analchem.0c02696 (2020).
- 14 Yan, X. *et al.* Temporal dynamics of cardiac immune cell accumulation following acute myocardial infarction. *J Mol Cell Cardiol* **62**, 24-35, doi:10.1016/j.yjmcc.2013.04.023 (2013).

Reviewers' comments:

Reviewer #1 (Remarks to the Author):

I would like to thank the authors for thoroughly revising their manuscript based on my comments. The authors addressed most of my comments. However, I am still not convinced and share the optimism of the authors that their Raman analysis approach can be seen as a “game changer”. Biomedical Raman spectroscopy Raman “lives and dies” with having the appropriate and tailored spectra analysis routines. However, the presented results do not convince me that the approach used by the authors will be a real game changer. They are mainly arguing that the spatial information has so far been used insufficiently and their approach is going far beyond machine learning approaches currently established. The results presented however do not show this. The authors even admit this by writing “We admit, that at the stage of this proof-of-concept-study the full potential of spatial analysis remains unclear”. That is exactly the point. Before the authors cannot convincingly show what their approach can do better as commonly applied machine learning approaches I don't recommend accepting this manuscript in such a high ranked journal than Nature communications. The authors claim that classical analysis methods need well-characterized training data sets limiting unbiased tissue characterization or identifying previously unknown features. If this would be shown on a statistically relevant large patient sample space accounting for intra- and inter patient variability I would be convinced and would recommend publication.

I also have an explanation why spatial information in Raman spectroscopy has been insufficiently used so far. This is mainly due to the slow scanning speed of Raman preventing the investigation of large tissue areas like is usually done for H&E and MALDI. For such small areas the spatial information is less pronounced as it would be for large areas scanned in spatial transcriptomics. Thus, the approach presented by the authors is probably only really useful for large sized samples. However, for such samples it would take extremely long to scan them with Raman spectroscopy. Furthermore, the correlation or co-registration between MALDI imaging and Raman microspectroscopy has been shown before in literature. Here the problem of the different information encoded within these modalities arises. MALDI spectra are much better resolved while within Raman spectra the detailed molecular information is often hidden under one Raman peak and cannot be spectrally resolved. Thus, a meaningful correlation between the rich MALDI peaks and the Raman peaks remains questionable.

Overall, the idea of using BayesSpace to fully integrate the spatial Raman information is definitely new and has to the best of my knowledge not been reported so far. However, the sample size area is in my opinion probably too small to show the full advantage of this approach. The data shown do not show the superiority of the presented approach to classical machine learning and the mentioned drawbacks of classical machine learning in terms of needing training data sets and not being able to identify unknown features is not shown. Thus, until the advantage of the presented approach is not convincingly shown I still cannot recommend publication in Nature communication.

Reviewer #2 (Remarks to the Author):

The revised manuscript addresses many of the concerns and suggestions raised during review.

The new Figure 6 begins to provide data that justifies many of the author's conclusions. It would be helpful to expand this analysis and provide more information on how well Raman spectroscopy delineates individual cell types found in the diseased myocardium. The current data could be better quantified to assess accuracy.

Reviewer #3 (Remarks to the Author):

The Authors have only partially addressed my concerns. I still believe the work, although novel and interesting, is too preliminary and it is therefore not possible to evaluate the potential impact of the proposed methodology for the study of experimental MI.

Rebuttal Letter

Reviewers' comments:

Reviewer #1 (Remarks to the Author):

I would like to thank the authors for thoroughly revising their manuscript based on my comments. The authors addressed most of my comments. However, I am still not convinced and share the optimism of the authors that their Raman analysis approach can be seen as a “game changer”. Biomedical Raman spectroscopy Raman “lives and dies” with having the appropriate and tailored spectra analysis routines. However, the presented results do not convince me that the approach used by the authors will be a real game changer. They are mainly arguing that the spatial information has so far been used insufficiently and their approach is going far beyond machine learning approaches currently established. The results presented however do not show this. The authors even admit this by writing “We admit, that at the stage of this proof-of-concept-study the full potential of spatial analysis remains unclear”. That is exactly the point. Before the authors cannot convincingly show what their approach can do better as commonly applied machine learning approaches I don't recommend accepting this manuscript in such a high ranked journal than Nature communications. The authors claim that classical analysis methods need well-characterized training data sets limiting unbiased tissue characterization or identifying previously unknown features. If this would be shown on a statistically relevant large patient sample space accounting for intra- and inter patient variability I would be convinced and would recommend publication.

Response:

We are grateful for the detailed and thoughtful feedback from the reviewer, which we believe have helped us again improving the quality of our work.

As this reviewer correctly points out, “biomedical Raman spectroscopy Raman “lives and dies” with having the appropriate and tailored spectra analysis routines.” So how can we achieve appropriate, reproducible, comparable, and tailored analytical routines? Most probably not by a colorful mix of different deep learning models and classical pipelines, all trained by different datasets and by different scientists with different focuses. Currently, there is no attempt to improve existing models – in contrast many research groups try to set up their own models. Hence, this will result in a great variety of poorly thought out, incomparable and barely reproducible analytical models. This important issue has been highlighted in recent articles¹⁻³.

Progress can only be made by referring to established pipelines, which are open-source and available to scientist of all fields. Analysis of Raman spectroscopy data would benefit far more when scientists of different specialties work together on the same model.

What is new about our approach is that we have tried to feed Raman data into existing analytical tools for single cell genomics. In contrast to existing analytical pipelines for Raman spectroscopy, tools in the latter field of research (single cell genomics) are currently exceedingly developed by scientists all over the world. Why shouldn't we benefit from this development and translate these advances to the field of Raman spectroscopy research?

We do not claim that our approach is superior to others, but it could be a “game changer” by its attempt for unifying and standardizing the analytical workflow to explore Raman data. Instead of simply reproducing the standard analyses on Raman data, we translated thoughtful concepts from (spatial) transcriptomics to the Raman world – like quality checks, pseudotime trajectories, spatially-aware clustering, or multimodal analyses.

However, we understand your concern that our manuscript may not have focused enough on the comparison with other machine learning methods, especially deep learning models. We agree that it is important to demonstrate the advantages of our approach over existing methods, and we have revised our manuscript to address this concern. Specifically, we have applied deep learning models to our current datasets, in order to compare the performance of our approach with that of 3 different deep learning models. Our additional findings indicate that clustering on the latent features extracted by autoencoder deep learning models only partially replicates the clusters identified by our method. To assess the degree of overlap, we measured the intersection over union between each cluster from “Raman Spectromics” cluster images in direct comparison to standard k-means clustering and latent deep learning on raw Raman spectra. The matrices beneath the cluster images reveal poor replication of Raman Spectromics clusters across other methods. On average, the intersection over union between each cluster image (shown in the swarmplot below) indicates that clustering on raw spectra by k-means or deep learning features can, at best, reproduce 60% of the clustering achieved by Raman Spectromics.

It is worth noting that the application of deep learning techniques to Raman data is often carried out with known labels for each spectrum, enabling the training of predictive models. However, since autoencoder models learn a reduced latent feature space by compressing and attempting to perfectly reconstruct the spectrum, there is no guarantee that these features will yield distinct clusters. In fact, in the literature, there is only one known example that applies an autoencoder to Raman data of cancer cell lines and visualizes the clustering of the latent features⁴.

Taken together, the data presented unequivocally highlight the substantial advantages of our method over deep learning approaches. Firstly, Raman Spectromics bypasses the need for developing and training deep learning models, eliminating the burdensome data requirements associated with such approaches. Secondly, as a heuristics-based method, Raman Spectromics provides superior interpretability, overcoming the inherent opacity and ‘black-box’ nature of deep learning methods. These advantages underscore the unique strengths of Raman Spectromics in terms of simplicity, data efficiency, and the generation of profound insights from the analysis.

Supplementary Fig. S17a

As we believe that these results are important and strengthen our work, we included it to new **Supplementary Fig. S17**.

Supplementary Fig. S17b

Comment:

I also have an explanation why spatial information in Raman spectroscopy has been insufficiently used so far. This is mainly due to the slow scanning speed of Raman preventing the investigation of large tissue areas like is usually done for H&E and MALDI. For such small areas the spatial information is less pronounced as it would be for large areas scanned in spatial transcriptomics. Thus, the approach presented by the authors is probably only really useful for large sized samples. However, for such samples it would take extremely long to scan them with Raman spectroscopy. Furthermore, the correlation or co-registration between MALDI imaging and Raman microspectroscopy has been shown before in literature. Here the problem of the different information encoded within these modalities arises. MALDI spectra are much better resolved while within Raman spectra the detailed molecular information is often hidden under one Raman peak and cannot be spectrally resolved. Thus, a meaningful correlation between the rich MALDI peaks and the Raman peaks remains questionable. Overall, the idea of using BayesSpace to fully integrate the spatial Raman information is definitely new and has to the best of my knowledge not been reported so far. However, the sample size area is in my opinion probably too small to show the full advantage of this approach.

Response:

We are thankful for this valuable feedback from the reviewer and understand his/her reasoning. However, we would like to clarify some points.

First, the reviewer states that “the correlation or co-registration between MALDI imaging and Raman microspectroscopy has been shown before”. Unfortunately, the reviewer made no reference. Indeed, there are article combining Raman and MALDI^{5,6}, however neither of those are taking respect of the spatial resolution of both scans or even perform cluster analyses for detecting myocardial compartments in infarcted hearts. Also, our intention was to show the feasibility of combining Raman data with other data modalities in the sense of multimodal analyses. And the data generated and especially the approach doing so by translating genomic tools is definitely novel. With the rest of the reviewer’s argumentation on correlation of MALDI and Raman peaks we fully agree.

The reviewer’s second point was about the size of the scan areas. We share the reviewer’s concerns about the scanning time of Raman spectroscopy. However, methods to accelerate scanning are currently being investigated (e.g. CARS, Coherent anti-Stokes Raman Spectroscopy^{7,8}). From our experience, when comparing Raman to MALDI scan times, MALDI is not faster than Raman spectroscopy. For the scan of the whole section presented below (new **Supplementary Fig. 7**), the MALDI scan with a resolution of 50 μm took around 2 hours, while the Raman scan with a resolution of 30 μm only took around 1.5 hours. It is mainly the high resolution of Raman scans which drags out the scan time (as mentioned in the previous rebuttal letter Raman spectroscopy required a scan time of around 1.5 hours for a scan of 250x250 μm but with a resolution of 1 μm).

Thirdly, the reviewer suggests that the advantages of a spatially-aware clustering algorithm – as provided by BayesSpace – could be more apparent when examining larger scan areas. This is a very interesting and important point. In response to the reviewer’s request, we have conducted additional experiments and analyses. Specifically, we created whole heart Raman scans at a lower resolution (30 μm), large area scans at higher resolution (10 μm) and high-resolution scans (1 μm) of the same section, and performed both spatially-unaware clustering

(Seurat) and spatially-aware clustering (BayesSpace) in the sections. The results indicate that especially in the larger space, spatially-aware clustering shows its benefits, in reducing artificial clustering, e.g. as seen from the background of the scan (new **Supplementary Fig. S7c vs. S7d**). However, when conducting scans over large areas, the heterogeneity of the expanded region tends to obscure or diminish the finer details that were initially discernible in the small, high-resolution area. Consequently, crucial information pertaining to the high-resolution region may be inadvertently overlooked or lost.

Hence, the right balance must be struck between high resolution and an appropriate scan area size to detect features that are detectable in particular at the subcellular level on one side, as well as features that particularly pronounced in the larger space.

While efforts are also made in digitally improving resolution⁹, one possibility to overcome this issue is to perform several high-resolution scans on the same section, to capture both details present on subcellular level, and also detect features only present when exploring the larger space (as shown in the new **Supplementary Fig. S7**).

The data shown do not show the superiority of the presented approach to classical machine learning and the mentioned drawbacks of classical machine learning in terms of needing training data sets and not being able to identify unknown features is not shown. Thus, until the advantage of the presented approach is not convincingly shown I still cannot recommend publication in Nature communication.

Response:

We are thankful for this feedback and hope that the new data presented clarified our position and arguments.

In addition to the previously demonstrated experimental comparison of performance of Raman Spectromics, classical k-means clustering and deep learning, we would like to take the opportunity to provide a detailed argumentative comparison of all methods. To address your concerns, this overview emphasizes the limitations, benefits, and features of commonly applied analytical routines for Raman spectroscopy, deep learning models and how our approach performs in comparison to those. We believe that this direct comparison will help to clearly see the unique advantages of each approach.

	Standard multi-dimensional analysis	Deep Learning	“Raman Spectromics”
Methods	PCA ¹⁰⁻¹³ , TCA ^{11,14} , PLS ^{15,16} , SVM ¹⁷	CNNs, RNNs, GANs ^{18,19}	PCA, t-SNE, UMAP, DDRTree, wnnUMAP
Analysis	Identification of differences between spectral signatures ^{20,21}	Additionally differentiation between healthy and diseased tissue ^{4,22,23}	Additionally quality check, visualization, spatially-aware cluster analysis, pseudotime trajectory analysis, spatial trajectory analysis, multimodal analysis
Standardization, comparability & reproducibility	Low (code is often not open-source; software is often chargeable)	Low to moderate ²⁴ (depends on if models are published open source; however deep learning models are often not further trained as soon as they fulfil their purpose)	High (code is open-source and under continuous development ²⁵⁻²⁷ ; the workflow documented online and supplemented by vignettes)
QC (quality check)	Low (no consistent standardized QC established)	Low (unless QC was performed before feeding data into the deep learning model)	High (QC provided by Seurat package and clustree package)
Dealing with inter- and intra-sample heterogeneity, Outlier “stability”	Low to moderate (no standardized workflow to detect outliers, artificial peaks etc.)	Low (no outlier detection except of done prior; alternatively a large sample cohort will be needed or model can be explicitly trained to handle outliers)	Moderate (includes outlier detection, paraffin removal and a robust backbone)
Improvability and development	Moderate (some advances for standard methods are made, e.g. spatially-aware PCA ²⁸)	Moderate to high (improvement of model only in case it is published open-source and is further trained)	High (all packages used are open-source and under continuous development; development ensured by a broad scientific community using the packages)
Data transfer within workflow and extensibility of workflow	Low (but possible by raw data)	Low to moderate (depending on the deep learning model trained; models can be trained for specific questions, or data can be used post-hoc for further analysis)	High (generated Seurat or sce objects can be easily transferred between packages; extension of workflow possible e.g., by multimodal analysis, weighted nearest neighbor analysis)

Reviewer #2 (Remarks to the Author):

The revised manuscript addresses many of the concerns and suggestions raised during review.

Response:

We are happy to hear your positive feedback concerning our revision and appreciate your acknowledgement of our additional work.

The new Figure 6 begins to provide data that justifies many of the author's conclusions. It would be helpful to expand this analysis and provide more information on how well Raman spectroscopy delineates individual cell types found in the diseased myocardium. The current data could be better quantified to assess accuracy.

Response:

We appreciate your recognition of the value of our new data shown, specifically on the new Figure 6. We fully agree with your important suggestion to expand the analysis and provide more information on how well Raman spectroscopy delineates individual cell types found in the diseased myocardium. In response to your feedback, we have expanded our analysis in the revised manuscript to include additional data and information regarding the differentiation of individual cell types using Raman spectroscopy.

We used the cell type-specific average spectra found in our previous analysis as “reference spectra” and calculate the similarity between the reference spectrum and each pixel spectrum of the scan. We used Euclidean distance as distance metric (**Supplementary Fig. S16**). Next, we compared the computationally identified cell types to the corresponding immunofluorescence staining as ground truth measure.

Subsequently we addressed your suggestion of quantifying the current data to assess accuracy. For this purpose, we have applied quantitative metrics such as sensitivity, specificity, and positive predictive value (from R package “caret”), to measure the performance of Raman spectroscopy in differentiating individual cell types (**Supplementary Fig. S16c**). These quantification measures provide an objective evaluation of the accuracy of our approach.

The analyzed data suggest that Raman spectroscopy has a high specificity but low sensitivity when detecting and delineating cell types. Except for erythrocytes, which have strong characteristic spectra and low heterogeneity, sensitivity was overall around 65%. Hence, Raman spectroscopy has a good potential to exclude the existence of specific cell types in a scan, but cannot securely say if these are in the scan.

However, one must consider important points when evaluating the capacities of differentiating cell types by Raman spectroscopy:

1. Previously we could show that Raman spectroscopy is an ultra-sensitive method, which could even discriminate different types of smooth muscle cells¹¹ or macrophages¹³. However, these experiments were conducted on isolated cells. Here, we try to identify cells in a tissue environment, and spectra obviously also contain information of the underlying or neighboring matrix.
2. Another point is abundance. In the present scan – as also shown in Fig. 6 h – immune cells only account to a minority of spectra in the scan. This is an important matter when evaluating sensitivity and specificity.

a

b

c

Reviewer #3 (Remarks to the Author):

The Authors have only partially addressed my concerns. I still believe the work, although novel and interesting, is too preliminary and it is therefore not possible to evaluate the potential impact of the proposed methodology for the study of experimental MI.

Response:

We thank the reviewer for his continued interest in our research work. In accordance with the comments of the editor and all reviewers, we have again substantially revised our manuscript to address the concerns raised.

We are pleased to read that you found our manuscript to be novel and interesting. Regarding your statement that “the work [...] is too preliminary to evaluate the potential impact [...] for the study of experimental MI, we would like to clarify our point of view.

The present work is not thought to explicitly study experimental MI. As the title says, this manuscript is about “Translating genomic tools to Raman spectroscopy analysis [to enable] high-dimensional tissue characterization on molecular levels”. Experimental MI is only used exemplarily, but we could also have investigated tumor biology or else. To investigate the applicability of Raman spectroscopy and its multidimensional, spatial capabilities, we employed two models of distinct myocardial pathologies (ischemia/reperfusion injury and Angiotensin II-induced myocardial hypertrophy) with well-known, well-characterized biological/histological features. Now that the complex bioinformatics workflow has been established and validated, upcoming in-depth analyses of hitherto unknown pathologies and disease mechanisms are made possible. Without a solid foundation, future analyses using our presented approach would be still questionable.

We are aware that for a thorough analysis of myocardial infarction this study remains superficial, but extending analysis explicitly for myocardial infarction would a) overflow the manuscript with irrelevant data and b) narrow our aim to address a broad readership which also includes bioinformaticians, biologist and non-cardiologist. Without doubt, a better understanding on the molecular patterns of ischemia/reperfusion by using Raman spectroscopy could be one of the most interesting aspects of our manuscript. However, this first requires a solid research platform which – once established – could be a steppingstone for upcoming research in this field.

In its current compact and concise form, the manuscript does not convey the extent of intensive, thorough, and comprehensive work we invested into the presented approach. We would like to ask the reviewer to accept that our manuscript still represents a valuable and novel methodological framework that will enable us and the broad scientific community to shed a novel light on both the method of Raman spectroscopy as well as creative, translational usage of well-known bioinformatic tools.

References

- 1 Boyuan Chen *et al.* Towards Training Reproducible Deep Learning Models. 2022 *IEEE/ACM 44th International Conference on Software Engineering (ICSE)*, doi:10.1145/3510003.3510163 (2022).
- 2 Heil, B. J. *et al.* Reproducibility standards for machine learning in the life sciences. *Nat Methods* **18**, 1132-1135, doi:10.1038/s41592-021-01256-7 (2021).
- 3 Hartley, M. & Olsson, T. S. G. dtoolAI: Reproducibility for Deep Learning. *Patterns (N Y)* **1**, 100073, doi:10.1016/j.patter.2020.100073 (2020).
- 4 He, C. *et al.* Accurate Tumor Subtype Detection with Raman Spectroscopy via Variational Autoencoder and Machine Learning. *ACS Omega* **7**, 10458-10468, doi:10.1021/acsomega.1c07263 (2022).
- 5 Ryabchykov, O., Popp, J. & Bocklitz, T. Fusion of MALDI Spectrometric Imaging and Raman Spectroscopic Data for the Analysis of Biological Samples. *Front Chem* **6**, 257, doi:10.3389/fchem.2018.00257 (2018).
- 6 Kirchberger-Tolstik, T. *et al.* Nondestructive molecular imaging by Raman spectroscopy vs. marker detection by MALDI IMS for an early diagnosis of HCC. *Analyst* **146**, 1239-1252, doi:10.1039/d0an01555e (2021).
- 7 Ozeki, Y. *et al.* High-speed molecular spectral imaging of tissue with stimulated Raman scattering. *Nature Photonics* **6**, 845-851, doi:10.1038/nphoton.2012.263 (2012).
- 8 Evans, C. L. *et al.* Chemical imaging of tissue in vivo with video-rate coherent anti-Stokes Raman scattering microscopy. *Proceedings of the National Academy of Sciences* **102**, 16807-16812, doi:10.1073/pnas.0508282102 (2005).
- 9 Horgan, C. C. *et al.* High-Throughput Molecular Imaging via Deep-Learning-Enabled Raman Spectroscopy. *Anal Chem* **93**, 15850-15860, doi:10.1021/acs.analchem.1c02178 (2021).
- 10 Marzi, J. *et al.* Marker-Independent In Situ Quantitative Assessment of Residual Cryoprotectants in Cardiac Tissues. *Anal Chem* **91**, 2266-2272, doi:10.1021/acs.analchem.8b04861 (2019).
- 11 Marzi, J., Brauchle, E. M., Schenke-Layland, K. & Rolle, M. W. Non-invasive functional molecular phenotyping of human smooth muscle cells utilized in cardiovascular tissue engineering. *Acta Biomater* **89**, 193-205, doi:10.1016/j.actbio.2019.03.026 (2019).
- 12 Zbinden, A. *et al.* Non-invasive marker-independent high content analysis of a microphysiological human pancreas-on-a-chip model. *Matrix Biol* **85-86**, 205-220, doi:10.1016/j.matbio.2019.06.008 (2020).
- 13 Feuerer, N. *et al.* Lipidome profiling with Raman microspectroscopy identifies macrophage response to surface topographies of implant materials. *Proc Natl Acad Sci U S A* **118**, doi:10.1073/pnas.2113694118 (2021).
- 14 Sugiyama, K. *et al.* Raman microspectroscopy and Raman imaging reveal biomarkers specific for thoracic aortic aneurysms. *Cell Rep Med* **2**, 100261, doi:10.1016/j.xcrm.2021.100261 (2021).
- 15 Goetz, M. J., Jr., Cote, G. L., Erckens, R., March, W. & Motamedi, M. Application of a multivariate technique to Raman spectra for quantification of body chemicals. *IEEE Trans Biomed Eng* **42**, 728-731, doi:10.1109/10.391172 (1995).
- 16 Hedegaard, M. *et al.* Discriminating isogenic cancer cells and identifying altered unsaturated fatty acid content as associated with metastasis status, using k-means clustering and partial least squares-discriminant analysis of Raman maps. *Anal Chem* **82**, 2797-2802, doi:10.1021/ac902717d (2010).
- 17 Widjaja, E., Zheng, W. & Huang, Z. Classification of colonic tissues using near-infrared Raman spectroscopy and support vector machines. *International Journal of Oncology*, doi:10.3892/ijo.32.3.653 (2008).
- 18 Luo, R., Popp, J. & Bocklitz, T. Deep Learning for Raman Spectroscopy: A Review. *Analytica* **3**, 287-301, doi:10.3390/analytica3030020 (2022).
- 19 Verbeeck, N., Caprioli, R. M. & Van de Plas, R. Unsupervised machine learning for exploratory data analysis in imaging mass spectrometry. *Mass Spectrom Rev* **39**, 245-291, doi:10.1002/mas.21602 (2020).

- 20 Ditta, A. *et al.* Principal components analysis of Raman spectral data for screening of Hepatitis C infection. *Spectrochim Acta A Mol Biomol Spectrosc* **221**, 117173, doi:10.1016/j.saa.2019.117173 (2019).
- 21 Guo, S., Rösch, P., Popp, J. & Bocklitz, T. Modified PCA and PLS: Towards a better classification in Raman spectroscopy-based biological applications. *Journal of Chemometrics* **34**, doi:10.1002/cem.3202 (2020).
- 22 Huang, L. *et al.* Rapid, label-free histopathological diagnosis of liver cancer based on Raman spectroscopy and deep learning. *Nat Commun* **14**, 48, doi:10.1038/s41467-022-35696-2 (2023).
- 23 Guleken, Z. *et al.* An application of raman spectroscopy in combination with machine learning to determine gastric cancer spectroscopy marker. *Comput Methods Programs Biomed* **234**, 107523, doi:10.1016/j.cmpb.2023.107523 (2023).
- 24 Liu, C. *et al.* On the Reproducibility and Replicability of Deep Learning in Software Engineering. *ACM Transactions on Software Engineering and Methodology* **31**, 1-46, doi:10.1145/3477535 (2021).
- 25 Satija, R. & Hoffmann, P. *SEURAT - R toolkit for single cell genomics*, accessed 05/22/2022).
- 26 Butler, A., Hoffman, P., Smibert, P., Papalexi, E. & Satija, R. Integrating single-cell transcriptomic data across different conditions, technologies, and species. *Nat Biotechnol* **36**, 411-420, doi:10.1038/nbt.4096 (2018).
- 27 Zhao, E. *et al.* Spatial transcriptomics at subspot resolution with BayesSpace. *Nat Biotechnol* **39**, 1375-1384, doi:10.1038/s41587-021-00935-2 (2021).
- 28 Shang, L. & Zhou, X. Spatially aware dimension reduction for spatial transcriptomics. *Nat Commun* **13**, 7203, doi:10.1038/s41467-022-34879-1 (2022).

REVIEWERS' COMMENTS

Reviewer #1 (Remarks to the Author):

The authors made a significant effort to address my concerns. I really appreciate the thoroughness with which they addressed my concerns and the new data they included. These new data are very nice. However, my concern (which is also confirmed by Referee #3) still remains that the manuscript is not significant enough for Nature Communications. The “game changer” potential the authors claim (in “unifying and standardizing the analytical workflow to explore Raman data”) has not really been proven. I appreciate that the authors have also applied deep learning approaches and I am not really surprised that the deep learning models only partially replicate the clusters identified by the authors’ method. Deep learning is data demanding and will show its potential only if a large amount of data is present. Thus, the game changer potential of deep learning has also not been shown due to the lack of large datasets. Thus, the comparison is also not really fair. The authors mention the black box character of deep learning. Here explainable AI should be mentioned which might help to overcome this drawback. However, the evaluation method (“feeding Raman data into existing analytical tools for single cell genomics”) presented by the authors is new, has some potential and the results presented are scientifically sound. Thus, I leave it up to the editor if the manuscript should be published in Nature communications.

Reviewer #2 (Remarks to the Author):

The authors have adequately responded to the comments and questions raised by this reviewer.

Reviewer #3 (Remarks to the Author):

Again, my concerns have been only partially addressed.

In my previous review, I highlighted that the data were too preliminary, and was therefore not possible to evaluate the potential impact of the proposed methodology for the study of experimental MI.

The authors have replied that "the present work is not thought to explicitly study experimental MI.....Experimental MI is only used exemplarily....We are aware that for a thorough analysis of myocardial infarction, this study remains superficial".

I fully understand that the aim of this study is to establish a solid research platform. But if the platform is not useful to address key biological questions, the overall impact remains debatable.

Rebuttal Letter

Reviewers' comments:

Reviewer #1 (Remarks to the Author):

The authors made a significant effort to address my concerns. I really appreciate the thoroughness with which they addressed my concerns and the new data they included. These new data are very nice. However, my concern (which is also confirmed by Referee #3) still remains that the manuscript is not significant enough for Nature Communications. The “game changer” potential the authors claim (in “unifying and standardizing the analytical workflow to explore Raman data”) has not really been proven.

I appreciate that the authors have also applied deep learning approaches and I am not really surprised that the deep learning models only partially replicates the clusters identified by the authors' method. Deep learning is data demanding and will show its potential only if a large amount of data is present. Thus, the game changer potential of deep learning has also not been shown due to the lack of large datasets. Thus, the comparison is also not really fair. The authors mention the black box character of deep learning. Here explainable AI should be mentioned which might help to overcome this drawback.

However, the evaluation method (“feeding Raman data into existing analytical tools for single cell genomics”) presented by the authors is new, has some potential and the results presented are scientifically sound. Thus, I leave it up to the editor if the manuscript should be published in Nature communications.

Response:

Once again, we are grateful for the detailed and thoughtful feedback from this reviewer, which has supported our determination in addressing this reviewer's concerns.

Reviewer #2 (Remarks to the Author):

The authors have adequately responded to the comments and questions raised by this reviewer.

Response:

We are thankful for this reviewer's feedback that has significantly helped improving the manuscript.

Reviewer #3 (Remarks to the Author):

Again, my concerns have been only partially addressed.

In my previous review, I highlighted that the data were too preliminary, and was therefore not possible to evaluate the potential impact of the proposed methodology for the study of experimental MI.

The authors have replied that "the present work is not thought to explicitly study experimental MI.....Experimental MI is only used exemplarily.... We are aware that for a thorough analysis of myocardial infarction, this study remains superficial".

I fully understand that the aim of this study is to establish a solid research platform. But if the platform is not useful to address key biological questions, the overall impact remains debatable.

Response:

We fully understand this author's concerns, but must also interject that the present manuscript is intended as a methods paper. Building on this, we will show the impact and significance in future projects. Incorporating data from those projects into the current manuscript would have blown up the content and reduced the generalizability of the presented approach.